# Invariance Learning in Deep Neural Networks with Differentiable Laplace Approximations

**Alexander Immer**[*,1,2]     **Tycho F.A. van der Ouderaa**[*,3]
**Gunnar Rätsch**[1]     **Vincent Fortuin**[1,4]     **Mark van der Wilk**[3]

[1]Department of Computer Science, ETH Zurich, Switzerland
[2]Max Planck Institute for Intelligent Systems, Tübingen, Germany
[3]Department of Computing, Imperial College London, UK
[4]Department of Engineering, University of Cambridge, UK

## Abstract

Data augmentation is commonly applied to improve performance of deep learning by enforcing the knowledge that certain transformations on the input preserve the output. Currently, the data augmentation parameters are chosen by human effort and costly cross-validation, which makes it cumbersome to apply to new datasets. We develop a convenient gradient-based method for selecting the data augmentation without validation data during training of a deep neural network. Our approach relies on phrasing data augmentation as an invariance in the prior distribution on the functions of a neural network, which allows us to learn it using Bayesian model selection. This has been shown to work in Gaussian processes, but not yet for deep neural networks. We propose a differentiable Kronecker-factored Laplace approximation to the marginal likelihood as our objective, which can be optimised without human supervision or validation data. We show that our method can successfully recover invariances present in the data, and that this improves generalisation and data efficiency on image datasets.

## 1 Introduction

Data augmentation is a commonly used machine learning technique that is essential to high-performing deep learning and computer vision systems. It aims to obtain a model that is *invariant* to a set or distribution of transformations, by fitting a model with inputs that are transformed in a way that is known to leave the output class unchanged. This procedure can be regarded as artificially creating more data and is well known to increase generalisation performance and data efficiency. Yet, choosing the right transformations is an expensive and task-specific process that relies on domain knowledge and human effort, as well as trial-and-error through cross-validation. This can quickly become intractable when many parameters are considered, particularly if they are continuous, because each setting requires training a model to convergence.

We aim to make selecting suitable transformations easier, by learning them via gradient descent. Our approach is inspired by the procedure of van der Wilk et al. (2018), which casts learning invariances and data augmentations as a Bayesian model selection problem. This view suggests selecting invariances by maximising the *marginal likelihood* with gradient-based optimisation. While this approach was successful in Gaussian process models, it has not yet been demonstrated in deep neural networks, where the marginal likelihood is harder to approximate.

---

*Equal contribution, order decided by coin flip. Correspondence to: `alexander.immer@inf.ethz.ch`, `tycho.vanderouderaa@imperial.ac.uk`

The code is available at `https://github.com/tychovdo/lila`

36th Conference on Neural Information Processing Systems (NeurIPS 2022).

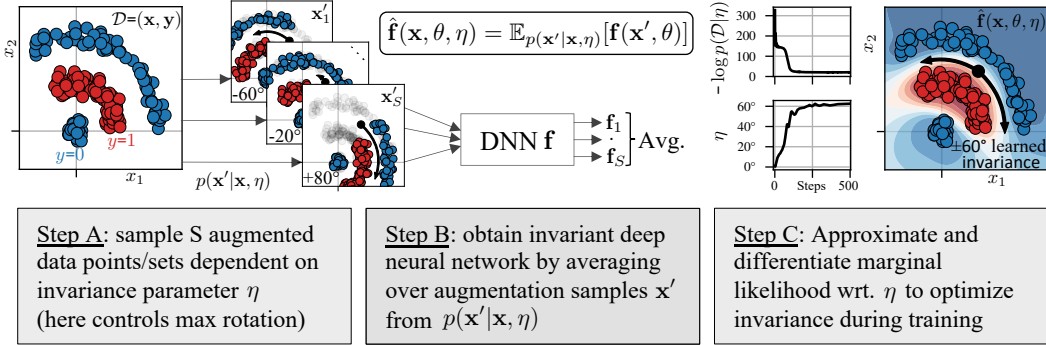

Step A: sample S augmented data points/sets dependent on invariance parameter $\eta$ (here controls max rotation)

Step B: obtain invariant deep neural network by averaging over augmentation samples $\mathbf{x}'$ from $p(\mathbf{x}'|\mathbf{x}, \eta)$

Step C: Approximate and differentiate marginal likelihood wrt. $\eta$ to optimize invariance during training

Figure 1: Illustration of our approach on a dataset with rotational invariance. Steps A and B allow to define a Bayesian neural network with a likelihood dependant on invariance parameter $\eta$ (van der Wilk et al., 2018). Our contributions enable tractable marginal likelihood estimation and differentiation during training, as illustrated in Step C. The right-most figure shows the posterior predictive of our Bayesian neural network after learning the invariance present in the data.

To circumvent this problem, we built upon the scalable Laplace approximation to the marginal likelihood developed by Immer et al. (2021a), who recently showed that maximising it can successfully select neural network hyperparameters, such as architectures. We extend their method to enable gradient-based optimisation of complex hyperparameters that control invariances in deep neural networks. To that end, we propose an efficient and differentiable Kronecker-factored Laplace approximation for invariant neural networks and a novel method to obtain stochastic gradients with respect to invariance parameters, which is also useful for optimising other hyperparameters. Our method is the first to enable differentiable Bayesian model selection to learn complex hyperparameters, in particular invariances, in deep neural networks.

Our approach is illustrated in Fig. 1. We specify invariances as a parameterised distribution over perturbations of the network's input, like in data augmentation (Step A). The model output is averaged over samples from the distribution (Step B), which yields a Bayesian neural network with likelihood dependent on the perturbations (Nabarro et al., 2021). We derive a marginal likelihood approximation for such neural networks and show how to efficiently compute its gradients with respect to the invariance parameters (Step C). By approximating the marginal likelihood, we can differentiably learn invariances during training, jointly with neural network parameters, and without validation data.

We demonstrate experimentally that our method can differentiably learn useful distributions over affine invariances, which are common data augmentations, on various versions of the image classification datasets MNIST, FashionMNIST, and CIFAR-10, without validation data. Our learned invariances improve the generalisation and data efficiency of neural networks, without the effort required for choosing data augmentations or custom architectures. On original datasets, our method can increase test performance by up to 8-10 percentage points. On random subsets of image classification datasets, we show that our method can achieve up to $10\times$ better data efficiency. Our work strengthens how Bayesian methods can be useful for deep learning beyond predictive uncertainty estimation.

## 2 Related Work

**Invariances in deep learning.** Since the inception of the convolutional neural network (CNN) (Fukushima and Miyake, 1982; LeCun et al., 1998), building invariances and equivariances into deep learning models has drastically increased data efficiency and generalisation (Cohen et al., 2018; Brandstetter et al., 2021), for instance, on image classification (Cohen and Welling, 2016), molecular dynamics (Batzner et al., 2021), and reinforcement learning (van der Pol et al., 2020). However, these approaches require knowing the invariances a priori. In this work, on the other hand, we aim to automatically learn the correct type and amount of invariance from data without supervision.

**Learning invariances.** Learning invariances from data is hard, because symmetries define constraints on the functions a network can represent, and therefore do not improve data fit according to the training loss, even if they would lead to better generalisation on test data. Some methods have therefore proposed to learn invariances or data augmentation by estimating gradients on the validation loss. For example, AutoAugment (Cubuk et al., 2018) learns data augmentation using

policy gradients, Lorraine et al. (2020) use the implicit function theorem, and Zhou et al. (2020) phrase the problem as meta-learning. Such approaches require validation sets of sufficient size to prevent high variance and overfitting (Lorraine et al., 2020). Here, we tackle the problem of learning invariances when validation data is not available.

**Validation-free invariance learning.** When casting invariance learning as a Bayesian model selection problem, we can select helpful invariances using the marginal likelihood of the model with training data alone. This has been successfully demonstrated in Gaussian processes (van der Wilk et al., 2018) and in the weight space of single-layer neural networks (van der Ouderaa and van der Wilk, 2021). To scale this approach to deep networks, Schwöbel et al. (2022) attempt to use a marginal likelihood that is only computed in the last layer. While the latter was successful for small neural networks, it failed to learn invariances on more complex datasets that require deep neural networks, likely due to known limitations of last-layer approaches (Ober et al., 2021; van Amersfoort et al., 2021). Augerino (Benton et al., 2020) is an alternative to Bayesian model selection that works for deep learning by regularising invariances to increase during training. However, this approach depends on the chosen parameterisation of the invariance and may still require a validation set to tune the regularisation strength. In App. C, we show that these issues can be detrimental to Augerino's performance. In contrast, our proposed method uses the Bayesian marginal likelihood estimate in deep neural networks that is parameterisation-invariant and improves performance. The main differences between our proposed method and the alternatives are summarised in App. A.

**Bayesian model selection for deep learning.** Marginal likelihood approximations have recently enabled gradient-based hyperparameter optimisation for deep neural networks (Immer et al., 2021a; Ober and Aitchison, 2021; Antorán et al., 2022a). Laplace approximations (MacKay, 1992) with structured Hessian approximations, such as KFAC (Martens and Grosse, 2015; Botev et al., 2017), or linearisation have been shown to improve generalisation, for example, by optimising weight regularisation (Immer et al., 2021a; Daxberger et al., 2021) and learning length-scales of convolutions (Antorán et al., 2022a). Other approximate inference methods that rely on ensembling or sampling do not directly apply to marginal likelihood estimation while common methods like mean-field variational inference (Blundell et al., 2015) or last-layer approaches (Ober et al., 2021; Daxberger et al., 2021; Schwöbel et al., 2022) have been shown to fail for model selection. We therefore focus on Laplace approximations and extend them to enable gradient-based optimisation of more complex hyperparameters like invariances.

## 3 Background

We consider a supervised learning task with dataset $\mathcal{D} = \{(\mathbf{x}_n, \mathbf{y}_n)\}_{n=1}^N$ consisting of $N$ inputs $\mathbf{x} \in \mathbb{R}^D$ and targets $\mathbf{y} \in \mathbb{R}^C$. Our goal is to learn the function $\mathbf{f} : \mathbb{R}^D \to \mathbb{R}^C$ that relates the inputs and outputs, which we represent as a neural network $\mathbf{f}(\mathbf{x}; \boldsymbol{\theta})$ with parameters $\boldsymbol{\theta}$. We control which solutions for $\mathbf{f}$ are preferred over others (i.e., the *inductive bias*) with hyperparameters $\mathcal{H}$, which parameterise the prior over $\boldsymbol{\theta}$ and $\mathbf{f}$. Invariance is a particularly helpful inductive bias that constrains the output of $\mathbf{f}$ to remain similar for certain transformations of the input $\mathbf{x}$, which can improve generalisation by allowing a single datapoint to inform predictions for a wider range of inputs. Our goal is to learn useful invariances together with the network weights.

### 3.1 Parameterising Invariance

To construct and parameterise invariant functions, we consider a local invariance that intuitively requires the function to not change "too much" in response to transformed inputs. To obtain such an invariant function $\hat{\mathbf{f}}$, we average an unconstrained function $\mathbf{f}$ over a perturbation distribution $p(\mathbf{x}'|\mathbf{x}, \boldsymbol{\eta})$. In practice, we sample $\mathbf{x}'$ by reparameterising $\boldsymbol{\epsilon} \sim p(\boldsymbol{\epsilon})$ with a differentiable function $\mathbf{g}$ and approximate the expectation with $S$ Monte Carlo samples $\boldsymbol{\epsilon}_1, \ldots, \boldsymbol{\epsilon}_S \overset{\text{iid}}{\sim} p(\boldsymbol{\epsilon})$ (van der Wilk et al., 2018; Benton et al., 2020):

$$\hat{\mathbf{f}}(\mathbf{x}; \boldsymbol{\theta}, \boldsymbol{\eta}) = \mathbb{E}_{p(\mathbf{x}'|\mathbf{x}, \boldsymbol{\eta})}[\mathbf{f}(\mathbf{x}'; \boldsymbol{\theta})] = \mathbb{E}_{p(\boldsymbol{\epsilon})}[\mathbf{f}(\mathbf{g}(\mathbf{x}, \boldsymbol{\epsilon}; \boldsymbol{\eta}); \boldsymbol{\theta})] \approx \tfrac{1}{S}\textstyle\sum_s \mathbf{f}(\mathbf{g}(\mathbf{x}, \boldsymbol{\epsilon}_s; \boldsymbol{\eta}); \boldsymbol{\theta}), \quad (1)$$

where $\hat{\mathbf{f}}$ is differentiable in the parameters $\boldsymbol{\eta}$ that control the perturbation distribution and therefore the invariance. When the perturbation distribution is uniform on the orbit of a group, we recover exact invariance in $\hat{\mathbf{f}}$ (Kondor, 2008; Ginsbourger et al., 2016). The perturbation distribution resembles data augmentation applied to $\mathbf{f}$ instead of the loss and is also used at test time ($\hat{\mathbf{f}}$ depends on it). Because the unconstrained function $\mathbf{f}$ is a neural network, we refer to $\hat{\mathbf{f}}$ as an **invariant neural network**.

Given that we can parameterise a suitable data augmentation distribution $p(\mathbf{x}'|\mathbf{x}, \boldsymbol{\eta})$, we can learn invariances in arbitrary domains or group structures. What parameterisation works best in practice remains a research question. In our experiments, we consider a combination of uniform distributions and corresponding parameters $\boldsymbol{\eta} \in \mathbb{R}^6$ over 6 generator matrices that define a probability density over the group of affine transformations, similar to Benton et al. (2020) and detailed in App. B.

Finding invariance parameters $\boldsymbol{\eta}$ is hard because $\hat{\mathbf{f}}$ is a constrained version of $\mathbf{f}$ and, especially for flexible models, this cannot improve the data fit according to the standard training loss (see also App. G.1). To overcome this, we propose to use Bayesian inference which provides a convenient framework to optimise invariance parameters with gradients during training without validation data.

### 3.2 Bayesian Model Selection

Bayesian inference prescribes how unknowns, such as invariance parameters, should be determined from data. To infer hyperparameters $\mathcal{H}$ from data, we are interested in the their posterior, $p(\mathcal{H}|\mathcal{D}) \propto p(\mathcal{D}|\mathcal{H})p(\mathcal{H})$ (MacKay, 2003). Because this posterior is intractable, type II maximum likelihood (ML-II) is often used instead, which obtains a point estimate $\mathcal{H}^*$ for the optimal hyperparameters according to the *marginal likelihood* $p(\mathcal{D}|\mathcal{H})$, which requires integration over the model parameters.

ML-II is routinely used in Gaussian processes (Rasmussen and Williams, 2006, § 5.2) and trades off model simplicity with data fit (Rasmussen and Ghahramani, 2001). There are also strong relations to quantities from other statistical frameworks, like cross-validation (Fong and Holmes, 2020), Minimum Description Length (Grünwald, 2007), and generalisation error bounds (Germain et al., 2016). Van der Wilk et al. (2018) showed its usefulness for selecting invariances, and elaborated on the mechanism by which it works. The main advantage of the marginal likelihood,

$$p(\mathcal{D}|\mathcal{H}) = \int p(\mathcal{D}|\boldsymbol{\theta}, \mathcal{H})p(\boldsymbol{\theta}|\mathcal{H})\, \mathrm{d}\boldsymbol{\theta} = \int \prod_{n=1}^{N} p(\mathbf{y}_n|\mathbf{f}(\mathbf{x}_n; \boldsymbol{\theta}), \mathcal{H})p(\boldsymbol{\theta}|\mathcal{H})\, \mathrm{d}\boldsymbol{\theta}, \tag{2}$$

is that it can be computed from the training data alone and optimised with gradients. In our work, the integration is over the neural network parameters and requires particularly scalable approximations.

### 3.3 Bayesian Model Selection for Deep Learning

Computing the marginal likelihood for neural networks involves intractable integrals. The Laplace approximation (Laplace, 1774; MacKay, 2003, § 27) offers a solution by approximating the log joint of the parameters and data, $p(\mathcal{D}, \boldsymbol{\theta}|\mathcal{H})$, with a second-order Taylor expansion around a mode $\boldsymbol{\theta}_*$:

$$\log p(\mathcal{D}|\mathcal{H}) \approx \log p(\mathcal{D}, \boldsymbol{\theta}_*|\mathcal{H}) - \tfrac{1}{2} \log \left| \tfrac{1}{2\pi} \mathbf{H}_{\boldsymbol{\theta}_*} \right|, \tag{3}$$

where the first term decomposes into the log likelihood, a sum over data points, and the log prior, both of which are cheap and easy to evaluate and correspond to a typical training loss evaluated at a mode $\boldsymbol{\theta}_*$. The second term depends on the log determinant of the log-joint Hessian at the same mode

$$\mathbf{H}_{\boldsymbol{\theta}_*} \stackrel{\text{def}}{=} -\nabla^2_{\boldsymbol{\theta}} \log p(\mathcal{D}, \boldsymbol{\theta}|\mathcal{H})|_{\boldsymbol{\theta}=\boldsymbol{\theta}_*} = \mathbf{H}^{\text{NLL}}_{\boldsymbol{\theta}_*} - \nabla^2_{\boldsymbol{\theta}} \log p(\boldsymbol{\theta}|\mathcal{H})|_{\boldsymbol{\theta}=\boldsymbol{\theta}_*}, \tag{4}$$

where $\mathbf{H}^{\text{NLL}}_{\boldsymbol{\theta}_*}$ denotes the Hessian of the negative log likelihood. This approach allows to estimate the marginal likelihood using a MAP estimate $\boldsymbol{\theta}_*$ of the weights and its local curvature $\mathbf{H}_{\boldsymbol{\theta}_*}$.

To circumvent the high cost of estimating the full Hessian, structured generalised Gauss-Newton (GGN) approximations are preferred for model selection in deep learning, as also in optimisation (Martens, 2020; Bottou, 2010). Immer et al. (2021a) recently demonstrated successful hyperparameter and architecture selection with such approximations. Further, they observe empirically that their algorithm does not require to be at a mode $\boldsymbol{\theta}_*$, which allows for interleaved gradient-based optimisation of parameters and hyperparameters during training. With the Jacobian matrix $\mathbf{J}_{\boldsymbol{\theta}}(\mathbf{x}) \stackrel{\text{def}}{=} \frac{\partial \mathbf{f}(\mathbf{x}; \boldsymbol{\theta})}{\partial \boldsymbol{\theta}} \in \mathbb{R}^{C \times P}$ of the network output given input $\mathbf{x}$ w.r.t. parameters, and Hessian of the log likelihood w.r.t. network outputs $\boldsymbol{\Lambda}(\mathbf{f}) = -\nabla^2_{\mathbf{f}} \log p(\mathbf{y}|\mathbf{f})$, the GGN simplifies the negative-log-likelihood-dependent term of the Hessian:

$$\mathbf{H}^{\text{NLL}}_{\boldsymbol{\theta}} \approx \mathbf{H}^{\text{GGN}}_{\boldsymbol{\theta}} \stackrel{\text{def}}{=} \sum_{n=1}^{N} \mathbf{J}_{\boldsymbol{\theta}}(\mathbf{x}_n)^{\mathsf{T}} \boldsymbol{\Lambda}(\mathbf{f}(\mathbf{x}_n; \boldsymbol{\theta})) \mathbf{J}_{\boldsymbol{\theta}}(\mathbf{x}_n). \tag{5}$$

Here, we assume that $\mathbf{f}$ forms the natural parameters of an exponential family likelihood function (Murphy, 2012). In classification, for example, $\mathbf{f}$ are the logits. We refer to the resulting approximations as Laplace-GGN. To overcome the still intractable quadratic size of $\mathbf{H}^{\text{GGN}}_{\boldsymbol{\theta}}$ in $P$, Immer et al. (2021a) use structured approximations like KFAC (Martens and Grosse, 2015). Cheaper approximations, such as diagonal ones, often compromise accuracy (App. F; Daxberger et al., 2021).

## 3.4 Kronecker-Factored Gauss-Newton Approximation (KFAC)

Kronecker-factored approximations to the Gauss-Newton, such as KFAC, are commonly used for Laplace approximations as they currently seem to provide the best known trade-off between performance and complexity (Ritter et al., 2018; Daxberger et al., 2021). KFAC is a block-diagonal approximation to the Gauss-Newton matrix $\mathbf{H}_{\boldsymbol{\theta}}^{\mathrm{GGN}}$ where each block corresponds to a layer in the neural network (Martens and Grosse, 2015; Botev et al., 2017). It is particularly efficient because each block is represented as two Kronecker factors instead of one dense matrix. For example, the GGN block of a layer with $D \times G$ parameters, that is, a fully-connected layer connecting $D$ to $G$ neurons, would have quadratic memory complexity $\mathcal{O}(D^2 G^2)$ while the corresponding Kronecker approximation of KFAC is in $\mathcal{O}(D^2 + G^2)$ and is therefore even tractable for wide neural networks.

Mathematically, KFAC approximates the GGN of the $l$th block of the neural network parameters by enforcing a Kronecker factorisation across data points. This would only be exact for a single data point $\mathbf{x}_n$, for which we can write the GGN block corresponding to the parameters of the $l$th layer as

$$\mathbf{H}_{l,n}^{\mathrm{GGN}} = [\mathbf{a}_{l,n} \otimes \mathbf{g}_{l,n}]\boldsymbol{\Lambda}_n[\mathbf{a}_{l,n} \otimes \mathbf{g}_{l,n}]^\mathsf{T} = [\mathbf{a}_{l,n}\mathbf{a}_{l,n}^\mathsf{T}] \otimes [\mathbf{g}_{l,n}\boldsymbol{\Lambda}_n\mathbf{g}_{l,n}^\mathsf{T}] \stackrel{\text{def}}{=} \mathbf{A}_{l,n} \otimes \mathbf{G}_{l,n}, \quad (6)$$

where $\mathbf{a}_{l,n} \in \mathbb{R}^{D_l}$ is the input to the $l$th layer for data point $\mathbf{x}_n$, $\mathbf{g}_{l,n} \in \mathbb{R}^{G_l \times C}$ is the transposed Jacobian of the network output with respect to the output of the $l$th layer for $\mathbf{x}_n$, and $\boldsymbol{\Lambda}_n = \boldsymbol{\Lambda}(\mathbf{f}(\mathbf{x}_n; \boldsymbol{\theta}))$. Thus, the factors in Eq. 6 are the Jacobian terms as in the GGN (Eq. 5) but only for the $l$th layer, i.e., $\mathbf{J}_{\boldsymbol{\theta}_l}(\mathbf{x}_n)^\mathsf{T} = \mathbf{a}_{l,n} \otimes \mathbf{g}_{l,n}$. KFAC then approximates the sum over $N$ data points by summing up the Kronecker factors individually instead of breaking the Kronecker-factored structure:

$$\mathbf{H}_l^{\mathrm{GGN}} = \sum_{n=1}^N [\mathbf{a}_{l,n}\mathbf{a}_{l,n}^\mathsf{T}] \otimes [\mathbf{g}_{l,n}\boldsymbol{\Lambda}_n\mathbf{g}_{l,n}^\mathsf{T}] \approx \frac{1}{N} \left[ \underbrace{\sum_{n=1}^N \mathbf{a}_{l,n}\mathbf{a}_{l,n}^\mathsf{T}}_{\stackrel{\text{def}}{=} \mathbf{A}_l} \right] \otimes \left[ \underbrace{\sum_{n=1}^N \mathbf{g}_{l,n}\boldsymbol{\Lambda}_n\mathbf{g}_{l,n}^\mathsf{T}}_{\stackrel{\text{def}}{=} \mathbf{G}_l} \right], \quad (7)$$

where $\mathbf{A}_l \in \mathbb{R}^{D_l \times D_l}$ and $\mathbf{G}_l \in \mathbb{R}^{G_l \times G_l}$ can be understood as the uncentered covariance over $N$ data points of the inputs to the $l$th layer and the Jacobians wrt. the output of the $l$th layer, respectively. The normalization by $\frac{1}{N}$ is necessary to account for the additional terms that arise from distributing the sum over the factors. For a single-layer model, i.e., a linear model, KFAC is exact (cf. App. F).

## 4 Invariance Learning with Differentiable Laplace Approximations

We propose a Laplace-GGN approximation to the marginal likelihood for invariant neural networks and enable gradient-based optimisation of their invariance parameters during training, without the use of validation data (see Fig. 1 for a high-level overview). In our approach, we integrate the augmentation distribution $p(\mathbf{x}'|\mathbf{x}, \boldsymbol{\eta})$ into a Bayesian neural network model such that the marginal likelihood directly depends on the invariance parameters $\boldsymbol{\eta} \in \mathcal{H}$. In particular, this is due to a modified likelihood function $p(\mathbf{y}|\hat{\mathbf{f}}(\mathbf{x}; \boldsymbol{\theta}, \boldsymbol{\eta}), \mathcal{H})$ (c.f., Nabarro et al., 2021). This model enables optimisation of the invariance parameters $\boldsymbol{\eta}$ using gradient ascent on the log marginal likelihood,

$$\boldsymbol{\eta} \leftarrow \boldsymbol{\eta} + \nabla_{\boldsymbol{\eta}} \log p(\mathcal{D}|\mathcal{H} = \{\boldsymbol{\eta}, \mathcal{M}\}), \quad (8)$$

where $\mathcal{M} = \mathcal{H} \setminus \{\boldsymbol{\eta}\}$ are remaining hyperparameters, such as regularisation strength or model architecture. However, the tractable Laplace-GGN and KFAC approximations are not available for invariant Bayesian neural networks, which prohibits straightforward application of the update in Eq. 8 using the methods described in Sec. 3.

In the following, we extend the Laplace-GGN and KFAC approximations to invariant neural networks (Secs. 4.1 and 4.2), which enables optimising the log marginal likelihood in parallel to the neural network parameters, as in Immer et al. (2021a). However, their algorithm has an intractable memory complexity for computing the gradients w.r.t. invariance parameters or other complex hyperparameters that act on the neural network function $\mathbf{f}$ directly. In practice, they only considered gradient-based optimisation of hyperparameters that act linearly on the Hessian $\mathbf{H}_{\boldsymbol{\theta}}$, for example regularisation strength and observation noise. In Sec. 4.3, we lift this constraint by proposing a method to obtain gradients w.r.t. complex hyperparameters without memory overhead. The final algorithm and a discussion of approximations for invariance learning are detailed in Apps. F and H.

### 4.1 Laplace-GGN for an Invariant Neural Network

To define the Laplace-GGN approximation to the log marginal likelihood, the hyperparameter objective, we need to extend the GGN to invariant neural networks. For an invariant neural network with

parameter estimate $\boldsymbol{\theta}_*$, the log marginal likelihood approximation is given by

$$\log p(\mathcal{D}|\boldsymbol{\eta}, \mathcal{M}) \approx \sum_{n=1}^{N} \log p(\mathbf{y}_n|\hat{\mathbf{f}}(\mathbf{x}_n; \boldsymbol{\theta}_*, \boldsymbol{\eta}), \mathcal{M}) + \log p(\boldsymbol{\theta}_*|\mathcal{M}) - \frac{1}{2}\log|\frac{1}{2\pi}\hat{\mathbf{H}}_{\boldsymbol{\theta}_*}^{\mathrm{GGN}}(\boldsymbol{\eta})|, \quad (9)$$

where the first two terms constitute the training loss of the invariant neural network corresponding to the log joint as in the vanilla Laplace approximation in Eq. 3 and the last term is the Gauss-Newton approximation $\hat{\mathbf{H}}_{\boldsymbol{\theta}_*}^{\mathrm{GGN}}$ of the invariant neural network $\hat{\mathbf{f}}$, which we derive below. The first and last term depend on the invariance parameter $\boldsymbol{\eta}$ that we want to learn and differentiate with respect to. We use $S$ Monte Carlo samples $\boldsymbol{\epsilon}_1, \ldots, \boldsymbol{\epsilon}_S \overset{\mathrm{iid}}{\sim} p(\boldsymbol{\epsilon})$ to estimate the invariant neural network $\hat{\mathbf{f}}$ as in Eq. 1.

**The log-likelihood** term is approximated with $S$ Monte Carlo samples, which leads to a lower bound,

$$\sum_{n=1}^{N} \log p(\mathbf{y}_n|\hat{\mathbf{f}}(\mathbf{x}_n; \boldsymbol{\theta}, \boldsymbol{\eta}), \mathcal{M}) \geq \sum_{n=1}^{N} \mathbb{E}_{\boldsymbol{\epsilon}_1, \ldots, \boldsymbol{\epsilon}_S}[\log p(\mathbf{y}_n|\frac{1}{S}\sum_s \mathbf{f}(\mathbf{g}(\mathbf{x}_n, \boldsymbol{\epsilon}_s; \boldsymbol{\eta}); \boldsymbol{\theta}), \mathcal{M})]$$

$$\approx \sum_{n=1}^{N} \log p(\mathbf{y}_n|\frac{1}{S}\sum_s \mathbf{f}(\mathbf{g}(\mathbf{x}_n, \boldsymbol{\epsilon}_s; \boldsymbol{\eta}); \boldsymbol{\theta}), \mathcal{M}), \quad (10)$$

which is due the concavity of the log likelihood in its natural parameter $\hat{\mathbf{f}}$ and Jensen's inequality as shown by Nabarro et al. (2021), Schwöbel et al. (2022), and detailed in App. F.1. The subsequent Monte Carlo approximation is then unbiased. Increasing $S$ leads to a tighter bound and improves the approximation. In practice, we sample $\boldsymbol{\epsilon}_s$ independently per data point $\mathbf{x}_n$ to reduce correlation. We can obtain a stochastic gradient w.r.t. $\boldsymbol{\eta}$ by sampling a mini-batch of $M \ll N$ data but it is not possible to batch over the $S$ augmentations that parameterise the likelihood function. The runtime and memory complexity are therefore increased by a factor of $S$. For a subset of data and $S \leq 100$ augmentations, the gradient w.r.t. $\boldsymbol{\eta}$ can be computed using backpropagation (Benton et al., 2020).

**The Gauss-Newton** can be derived from the log-likelihood approximation in Eq. 10 using the same $S$ samples. In particular, the Jacobian and log-likelihood Hessian required for the GGN are given by

$$\hat{\mathbf{J}}_{\boldsymbol{\theta}}(\mathbf{x}; \boldsymbol{\eta}) \overset{\mathrm{def}}{=} \frac{1}{S}\sum_s \mathbf{J}_{\boldsymbol{\theta}}(\mathbf{g}(\mathbf{x}, \boldsymbol{\epsilon}_s; \boldsymbol{\eta})) \quad \text{and} \quad \hat{\boldsymbol{\Lambda}}(\mathbf{x}; \boldsymbol{\theta}, \boldsymbol{\eta}) \overset{\mathrm{def}}{=} \boldsymbol{\Lambda}(\frac{1}{S}\sum_s \mathbf{f}(\mathbf{g}(\mathbf{x}, \boldsymbol{\epsilon}_s; \boldsymbol{\eta}); \boldsymbol{\theta})). \quad (11)$$

Both terms depend on the invariance parameter $\boldsymbol{\eta}$ and are later differentiated with respect to it. The resulting GGN of the invariant neural network log likelihood estimated with $S$ samples is defined as

$$\hat{\mathbf{H}}_{\boldsymbol{\theta}}^{\mathrm{GGN}}(\boldsymbol{\eta}) \overset{\mathrm{def}}{=} \sum_{n=1}^{N} \hat{\mathbf{J}}_{\boldsymbol{\theta}}(\mathbf{x}_n; \boldsymbol{\eta})^{\mathsf{T}} \hat{\boldsymbol{\Lambda}}(\mathbf{x}_n; \boldsymbol{\theta}, \boldsymbol{\eta}) \hat{\mathbf{J}}_{\boldsymbol{\theta}}(\mathbf{x}_n; \boldsymbol{\eta}). \quad (12)$$

The GGN for an invariant model therefore requires averaging individual Jacobians and functions of the underlying neural network $\mathbf{f}$. This is in contrast to the GGN of an improper Bayesian model (Wenzel et al., 2020) with standard data augmentation, which averages log-likelihood terms instead of functions. In this case, entire GGN terms are averaged over the $S$ augmentations and the marginal likelihood cannot be optimised because it requires tempering (Immer et al., 2021a). Computing GGN of an invariant model increases the runtime by a factor of $S$ over a non-invariant model. The empirical Fisher, which is often cheaper, can be extended analogously and requires averaging gradients instead of Jacobians. Finally, the Laplace-GGN is obtained by computing the log determinant.

The Laplace-GGN approximation derived here already allows to learn small invariant neural networks with few layers and small datasets via automatic differentiation. For example, this is tractable for the classification example illustrated in Fig. 1. However, *computing* the log determinant of the GGN has a cubic complexity in the number of parameters $P$ and is therefore intractable for deep neural networks on larger datasets. To enable its estimation, we extend KFAC to invariant neural networks in Sec. 4.2. Further, *differentiating* the log determinant has intractable memory complexity because it does not allow for a stochastic gradient but requires construction of a computational graph for the entire GGN approximation. This is a key limitation of the method proposed by Immer et al. (2021a) when optimising complex hyperparameters, for which we provide a solution in Sec. 4.3.

## 4.2 Extending KFAC to Invariant Neural Networks

Computing KFAC as described in Sec. 3 for an invariant neural network would not preserve the Kronecker structure and therefore be intractable for deep neural networks due to the quadratic cost in the numbers of parameters per layer. In particular in Eq. 6, KFAC uses the fact that the Jacobian w.r.t. the parameters of the $l$th layer, $\mathbf{J}_{\boldsymbol{\theta}_l}(\mathbf{x}_n)$, can be written as the Kronecker product $\mathbf{a}_{l,n} \otimes \mathbf{g}_{l,n}$. For an invariant neural network, the Kronecker structure cannot be maintained because of the sum over $S$ augmentation-sample Jacobians, each of which constitutes a Kronecker product:

$$\hat{\mathbf{J}}_{\boldsymbol{\theta}_l}(\mathbf{x}_n; \boldsymbol{\eta}) = \frac{1}{S}\sum_s \mathbf{J}_{\boldsymbol{\theta}_l}(\mathbf{g}(\mathbf{x}_n, \boldsymbol{\epsilon}_s; \boldsymbol{\eta})) = \frac{1}{S}\sum_s [\mathbf{a}_{l,n}^{(s)} \otimes \mathbf{g}_{l,n}^{(s)}], \quad (13)$$

where $\mathbf{a}^{(s)}, \mathbf{g}^{(s)}$ depend on the $s$th sample $\mathbf{x}_s = \mathbf{g}(\mathbf{x}, \boldsymbol{\epsilon}_s; \boldsymbol{\eta})$ and thus $\boldsymbol{\eta}$. In general, the sum of Kronecker products does not allow for efficient computation and requires evaluating the products, which is intractable as it requires $\mathcal{O}(D_l^2 G_l^2)$ instead of $\mathcal{O}(D_l^2 + G_l^2)$ memory.

Similar to the idea underlying KFAC itself (Martens and Grosse, 2015), we enforce the efficient Kronecker-factored structure by approximating the sum of a Kronecker product as a Kronecker product of sums and appropriately normalizing by the number of terms. However, instead of applying this approximation to the GGN over multiple data points, we apply it to the Jacobian and have

$$\hat{\mathbf{J}}_{\boldsymbol{\theta}_l}(\mathbf{x}_n; \boldsymbol{\eta}) = \tfrac{1}{S} \textstyle\sum_s [\mathbf{a}_{l,n}^{(s)} \otimes \mathbf{g}_{l,n}^{(s)}] \approx \tfrac{1}{S^2} [\textstyle\sum_s \mathbf{a}_{l,n}^{(s)}] \otimes [\textstyle\sum_s \mathbf{g}_{l,n}^{(s)}]. \tag{14}$$

Applying this Jacobian approximation, the $n$th summand of the GGN can be written as

$$\hat{\mathbf{H}}_{l,n}^{\mathrm{GGN}}(\boldsymbol{\eta}) \approx \left[ \tfrac{1}{S^2} [\textstyle\sum_s \mathbf{a}_{l,n}^{(s)}] \otimes [\textstyle\sum_s \mathbf{g}_{l,n}^{(s)}] \right] \hat{\boldsymbol{\Lambda}}(\mathbf{x}_n; \boldsymbol{\theta}, \boldsymbol{\eta}) \left[ \tfrac{1}{S^2} [\textstyle\sum_s \mathbf{a}_{l,n}^{(s)}] \otimes [\textstyle\sum_s \mathbf{g}_{l,n}^{(s)}] \right]^{\mathsf{T}} \tag{15}$$

$$= \tfrac{1}{S^4} \left[ [\textstyle\sum_s \mathbf{a}_{l,n}^{(s)}] [\textstyle\sum_s \mathbf{a}_{l,n}^{(s)}]^{\mathsf{T}} \right] \otimes \left[ [\textstyle\sum_s \mathbf{g}_{l,n}^{(s)}] \hat{\boldsymbol{\Lambda}}(\mathbf{x}_n; \boldsymbol{\theta}, \boldsymbol{\eta}) [\textstyle\sum_s \mathbf{g}_{l,n}^{(s)}]^{\mathsf{T}} \right] \stackrel{\mathrm{def}}{=} \hat{\mathbf{A}}_{l,n} \otimes \hat{\mathbf{G}}_{l,n}.$$

To compute the full KFAC, we then accumulate the Kronecker factors for the $l$th layer, $\hat{\mathbf{A}}_{l,n}$ and $\hat{\mathbf{G}}_{l,n}$, of the invariant neural network over all $n$ data points to obtain $\hat{\mathbf{A}}_l$ and $\hat{\mathbf{G}}_l$ as in Eq. 7 for vanilla KFAC. Like KFAC, our approximation for invariant models remains exact for linear models (App. F.1).

The log determinant required for the marginal-likelihood approximation can be computed from $\hat{\mathbf{G}}_l$, $\hat{\mathbf{A}}_l$ individually (Immer et al., 2021a) and has a complexity of $\mathcal{O}(D_l^3 + G_l^3)$ as opposed to the intractable $\mathcal{O}(D_l^3 G_l^3)$. This enables us to apply our method to deep invariant neural networks with widths of order $10^4$, like vanilla KFAC. We discuss computational complexities in depth in App. D. We note that KFAC for invariant neural networks could further be of independent interest for second-order optimisation and inference (Martens and Grosse, 2015; Zhang et al., 2018).

### 4.3 Efficient Gradient Estimation of the Laplace-GGN w.r.t. Complex Hyperparameters

Automatic differentiation (AD) of the log determinant term in the Laplace-GGN w.r.t. complex hyperparameters, such as invariances, has an intractable memory complexity. Here, we propose a method to estimate the gradient of the log determinant in the Laplace-GGN without memory overhead. For AD, the *memory* complexity is equivalent to the *runtime* complexity of computing the log determinant, which is at least $\mathcal{O}(NP)$, the cost of training a deep neural network for one epoch. Such computation is only tractable due to batching (e.g., for standard training losses) and otherwise would require several terabytes of memory for deep neural networks. However, the log determinant does not allow for a batched gradient and therefore AD requires storing the full training data pass.

Our approach to reducing the memory complexity relies on computing a vector-Jacobian product where both, the vector and the Jacobian, can be estimated from batches of data. Mathematically, the problem reduces to differentiation of the log determinant of a sum of square matrices w.r.t. hyperparameter $\boldsymbol{\eta}$, i.e., $\frac{\partial}{\partial \boldsymbol{\eta}} \log |\mathbf{H}(\boldsymbol{\eta})|$ with $\mathbf{H}(\boldsymbol{\eta}) = \sum_n \mathbf{H}_n(\boldsymbol{\eta}) \in \mathbb{R}^{P \times P}$. For a positive definite matrix $\mathbf{H}$, we have $\frac{\partial \log |\mathbf{H}|}{\partial \mathbf{H}} = \mathbf{H}^{-1}$. Therefore, we can differentiate w.r.t. $\boldsymbol{\eta}$ with

$$\frac{\partial}{\partial \boldsymbol{\eta}} \log |\mathbf{H}(\boldsymbol{\eta})| = \textstyle\sum_{p=1}^P \sum_{q=1}^P [\mathbf{H}^{-1}(\boldsymbol{\eta})]_{p,q} [\frac{\partial}{\partial \boldsymbol{\eta}} \sum_{n=1}^N \mathbf{H}_n(\boldsymbol{\eta})]_{p,q}$$

$$= \mathrm{vec}\left(\mathbf{H}^{-1}(\boldsymbol{\eta})\right)^{\mathsf{T}} \textstyle\sum_{n=1}^N \frac{\partial}{\partial \boldsymbol{\eta}} \mathrm{vec}\left(\mathbf{H}_n(\boldsymbol{\eta})\right), \tag{16}$$

where the two vectorised matrices are $P^2$-dimensional vectors and can both be computed from individual batches of $\mathbf{H}_n$ as follows: the first term acts as a *preconditioner* and can be computed by summing up the batches and inverting the resulting matrix $\mathbf{H}$ without storing the computation graph. The second term is a sum over $N$ Jacobians w.r.t. $\boldsymbol{\eta}$ and we can either aggregate it or obtain an unbiased stochastic estimate from batches of data. The product of both terms constitutes a vector-Jacobian product and is a standard procedure of AD (Paszke et al., 2017).

The proposed method allows to aggregate gradients with respect to complex hyperparameters with memory complexity that is controlled by the batch size $1 \leq M \leq N$. In contrast to naive application of AD to the log determinant, this allows to decouple memory and runtime complexity and enables gradient-based optimisation of the log determinant in Eq. 9 w.r.t. the augmentation parameters for deep invariant networks on large datasets. More generally, our method enables gradient-based marginal likelihood optimisation for more complex hyperparameters than previously considered (Immer et al., 2021a; Antorán et al., 2022a). In App. E, we describe the gradient computation for KFAC in detail.

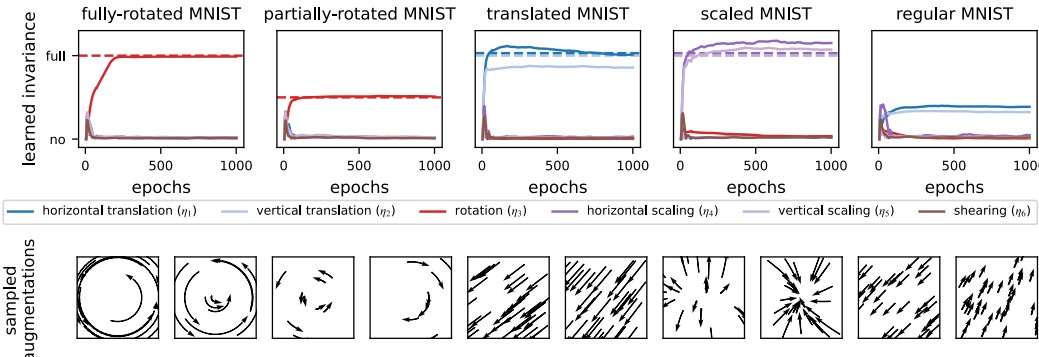

Figure 2: Trajectories of invariance parameters $\boldsymbol{\eta} = [\eta_1, \eta_2, ..., \eta_6]^T$ optimised with LILA over 1000 epochs for different versions of the MNIST dataset. From each distribution, two sampled transformations are represented as flow fields with arrow lengths scaled to pixel movement. LILA automatically learns the correct type and amount of invariance for each of the modified training datasets and keeps the other affine invariance parameters at zero as desired.

## 5  Experiments

We evaluate our method that *learns invariances using Laplace approximations* (LILA) by optimising affine invariances on different MNIST (LeCun and Cortes, 2010), FashionMNIST (Xiao et al., 2017), and CIFAR-10 (Krizhevsky et al., 2009) classification tasks. To validate whether the method is capable of learning appropriate invariances, we construct several additional datasets modified by known sets of invariance transformations with the goal to recover them. We consider the following affine transformations: full rotation, partial rotation, translation, and scaling (full details in App. B). We compare our approach with a non-invariant baseline and Augerino (Benton et al., 2020), which is, to our knowledge, the only other method that is capable of learning invariances on complex image datasets in deep neural networks without validation data. In the non-invariant model, prior parameters are learned using the marginal likelihood (Immer et al., 2021a). For invariance learning, prior parameters and $\boldsymbol{\eta}$ are jointly learned based on the marginal likelihood. For Augerino, we minimise the regular cross entropy with added regularising term $-10^{-2}||\boldsymbol{\eta}||_2$ and a fixed weight decay of $10^{-4}$, following the original paper (Benton et al., 2020). The same parameterisation, network architecture, and initialisations (in particular $\boldsymbol{\eta} = \mathbf{0}$) were used for all methods. We assess performance of our approach by inspecting learned invariances $\boldsymbol{\eta}$, marginal likelihoods, and final test performances.

### 5.1  Recovering Known Invariances

To assess the invariances learned by our method LILA, we can inspect the learned invariance parameters. The invariance parameter vector $\boldsymbol{\eta} = [\eta_1, \eta_2, ..., \eta_6]^T$ describes affine invariances with components corresponding to x-translation, y-translation, rotation, horizontal and vertical scaling, and shearing (App. B). As an MLP has little symmetry encoded in the architecture itself, we expect $\boldsymbol{\eta}$ to almost correctly recover the invariances. In Fig. 2, we plot the trajectories of each vector component over the course of training for an MLP model on different transformed MNIST datasets. As reference, we show the amount of invariance that was imposed on each dataset as a dashed line. From the figure, we can observe that for each dataset, LILA learns the correct invariance as well as the amount of each invariance. To some extent, the model also learned translational invariance on the regular MNIST dataset which can be explained by intrinsic translational invariance of the dataset.

Other network architectures have certain symmetries already built-in to some extent (e.g., translational equivariance of convolutional layers). Yet, we find that LILA is also capable of inferring the correct invariances with such larger and other network architectures and on a variety of datasets in App. J.3. In App. F, we further show that cheaper Hessian approximations, such as the diagonal GGN instead of KFAC and empirical Fisher (EF) instead of GGN lead to worse performance for invariance learning. This suggests that our extension of KFAC to invariant neural networks is necessary for sufficient invariance learning. While Immer et al. (2021a) find that diagonal approximations can suffice for learning regularisation hyperparameters, this does not apply for the more complex invariance parameters considered here and more accurate approximations tend to increase performance.

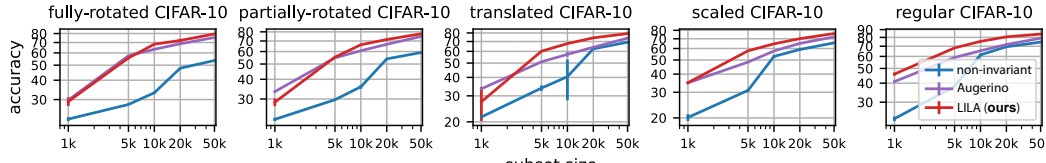

Figure 3: LILA improves data efficiency on different versions of CIFAR-10 by learning invariances. We compare test accuracy on randomly sampled subsets to a non-invariant network and Augerino using a ResNet. For example on fully-rotated CIFAR-10, invariance learning achieves the final accuracy of the non-invariant model with 10 times less data. In most cases, LILA further improves over Augerino. We plot the average performance and standard error over three seeds.

| Dataset | Network | Method | Fully Rotated | Partially Rotated | Translated | Scaled | Original |
|---|---|---|---|---|---|---|---|
| MNIST | MLP | non-invariant | 93.82 ±0.10 | 95.83 ±0.03 | 94.15 ±0.02 | 97.07 ±0.06 | 98.20 ±0.03 |
| | | Augerino | **97.83** ±0.03 | 96.35 ±0.02 | 94.47 ±0.08 | 97.45 ±0.03 | 98.45 ±0.03 |
| | | Diff. Laplace (**ours**) | 97.74 ±0.07 | **97.81** ±0.11 | **97.28** ±0.05 | **98.33** ±0.05 | **98.98** ±0.05 |
| | CNN | non-invariant | 95.97 ±0.33 | 97.51 ±0.17 | 96.54 ±0.29 | 98.37 ±0.00 | 99.09 ±0.02 |
| | | Augerino | **99.04** ±0.02 | 98.91 ±0.03 | 97.79 ±0.09 | 98.77 ±0.06 | 98.26 ±0.10 |
| | | LIL KFAC (**ours**) | 98.83 ±0.07 | **98.92** ±0.05 | **98.69** ±0.07 | **99.01** ±0.06 | **99.42** ±0.02 |
| F-MNIST | MLP | non-invariant | 77.62 ±0.30 | 81.10 ±0.23 | 77.68 ±0.10 | 81.84 ±0.05 | 88.48 ±0.56 |
| | | Augerino | 77.76 ±0.15 | 81.40 ±0.05 | 78.05 ±0.10 | 82.46 ±0.09 | 89.10 ±0.13 |
| | | LILA (**ours**) | **87.39** ±0.06 | **86.72** ±0.08 | **84.62** ±0.09 | **84.31** ±0.06 | **89.94** ±0.12 |
| | CNN | non-invariant | 78.69 ±0.28 | 82.12 ±0.35 | 80.33 ±0.19 | 83.66 ±0.37 | 89.54 ±0.23 |
| | | Augerino | 85.76 ±3.23 | 81.54 ±0.19 | 82.94 ±0.13 | 83.58 ±0.08 | 90.07 ±0.12 |
| | | LILA (**ours**) | **89.45** ±0.03 | **88.40** ±0.00 | **87.73** ±0.20 | **87.33** ±0.00 | **91.92** ±0.21 |
| CIFAR-10 | ResNet | non-invariant | 54.16 ±0.40 | 59.90 ±0.12 | 69.65 ±0.16 | 66.06 ±0.13 | 74.13 ±0.51 |
| | | Augerino | 75.40 ±0.19 | 74.76 ±0.34 | 73.71 ±0.31 | 72.07 ±0.09 | 79.03 ±1.04 |
| | | LILA (**ours**) | **79.50** ±0.62 | **77.71** ±0.46 | **79.21** ±0.17 | **76.03** ±0.15 | **84.19** ±0.76 |

Table 1: Test accuracy for models using LILA on different versions of the MNIST, FashionMNIST, and CIFAR-10 datasets. We report the average performance and the standard error over three random seeds. Our method outperforms the non-invariant network and Augerino for most models and datasets.

## 5.2 Invariance Learning in Different Networks

To quantify the benefit of models with learned invariances, we present final test accuracies of LILA and the baselines with different models on each of the datasets in Table 1. Additional marginal likelihood scores can be found in App. J.2. In terms of test accuracy and marginal likelihood, we find that learning invariances always outperforms the non-invariant baseline and that our approach improves over Augerino in almost all cases. This holds across the transformed and original datasets. While the improvements are modest for MNIST, the performance

| Method | Test accuracy |
|---|---|
| non-invariant | 85.17 ±0.39 |
| Augerino | 87.67 ±0.08 |
| LILA (**ours**) | **91.98** ±0.04 |
| LILA EF (**ours**) | 91.64 ±0.26 |

Table 2: Invariance Learning with KFAC on CIFAR-10 with a Wide ResNet achieves best test accuracy. Also, the cheaper empirical Fisher (EF) variant improves over Augerino.

improvements on F-MNIST and CIFAR-10 can be up to 10 percent points. In Table 2, we use the commonly used Wide ResNet architecture on CIFAR-10 and find that invariance learning with Augerino merely improves performance while our method achieves performance improvement of almost 7% points. We also report the performance of LILA with the cheaper KFAC-EF instead of GGN.

## 5.3 Invariance Learning Improves Data Efficiency

Invariances can be particularly useful for data-efficiency, which can be evaluated by measuring performance on subsets of data. In Fig. 3, we show the test accuracy of LILA, the non-invariant baseline, and Augerino trained on different subsets of CIFAR-10 and its modified versions. We further provide results for all architectures and datasets in App. J.5. In general, we find that invariance learning with both Augerino and our method always improves performance across datasets and subset sizes. Most notably on a subset of 1000 regular CIFAR-10 dataset samples, Augerino improved performance by 18 percentage points compared to the non-invariant baseline, whereas our method showed an improvement of 22 percentage points. LILA requires only 10% of the data to obtain the same accuracy as the non-invariant model on the fully-rotated dataset. These findings suggest that learning invariances is useful in general, and in particular when limited data is available.

### 5.4 On the Learned Distributions

In some instances (see CIFAR-10 results in Apps. J.3 and J.4), we observed that the model learned translational invariance on the scaled dataset rather than scale invariance. Our independent uniform distributions over invariance components cannot capture correlations between components whereas the scaled dataset was jointly scaled across the horizontal and vertical axes and thus is correlated. We hypothesise that more complex distributions that do allow for capturing correlations could offer a potential solution in such cases, but leave investigation of more complex families to future work.

## 6 Discussion and Limitations

We discuss the runtime complexity and approximations of our method in general, and for invariance learning in particular, in detail in App. D and App. F, respectively. The runtime complexity of LILA, just like that of Augerino and other methods that use test-time data augmentation, increases linearly by the factor $S$, which denotes the number of augmentation samples used. Since we know that we are sampling from a lower bound (Eq. 24), more samples are generally better and are expected to improve the performance, which we also observed in our experiments. In addition, LILA requires estimation and differentiation of the log-determinant term. While our extension of KFAC to sampling-based invariant models and the batched gradients make such computation at all tractable, it can still be expensive for a large number of classes $C$ and augmentation samples $S$. Using the EF instead of the GGN overcomes scaling in $C$ but is a cruder approximation. Although these additional computations make LILA slower than Augerino, they enable overcoming its issues, i.e., parameterisation-dependence and additional hyperparameters that need to be tuned (App. C), enable to learn soft invariances much more preciselly.

## 7 Conclusion

We presented a method that enables automatic invariance learning in deep neural networks directly from training data, without requiring supervision or validation data. The approach is inspired by using the marginal likelihood, which is a parameterisation-independent quantity coinciding with generalisation performance. To make this practical, we use a differentiable Laplace approximation to allow for gradient-based optimisation of invariances in deep learning. While the accuracy of the approximation is difficult to verify, we do show experimentally that the method is capable of learning invariances in MNIST, FashionMNIST, and CIFAR-10 datasets, leading to better marginal likelihoods and higher test performances. Our work shows that approximate Bayesian inference methods can be useful for learning complex hyperparameters, even in deep learning, and are therefore relevant beyond predictive uncertainty estimation. In future work, it would be interesting to improve the scalability and accuracy of marginal likelihood approximations which could enable learning even more complex hyperparameters, such as augmentation distributions parameterised by neural networks. Alternatively, improving parameterisations of invariances in neural networks could greatly improve the scalability of LILA and related approaches.

### Acknowledgements

A.I. acknowledges funding by the Max Planck ETH Center for Learning Systems (CLS). V.F. acknowledges funding by the Swiss Data Science Center through a PhD Fellowship, the Swiss National Science Foundation through a Postdoc.Mobility Fellowship, St John's College Cambridge through a Research Fellowship, and the Branco Weiss Foundation through a Branco Weiss Fellowship.

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
