## A  Comparison with Related Approaches

Table 3: Overview of related approaches and our proposed method. Note that only Augerino (Benton et al., 2020) and our method work for deep neural networks, while using only training data in a single run. Our method has the additional benefit of being Bayesian, parameterisation-independent, and requires no additional hyperparameter, which empirically can lead to better performance. We discuss the corresponding issues of Augerino in detail in App. C

| Approach | single run | train data only | deep neural network | Bayesian justification | parameterisation-independent |
|---|:---:|:---:|:---:|:---:|:---:|
| Cubuk et al. (2018) | ✗ | ✗ | ✓ | ✗ | ✓ |
| Zhou et al. (2020) | ✗ | ✗ | ✓ | ✗ | ✓ |
| Lorraine et al. (2020) | ✗ | ✗ | ✓ | ✗ | ✓ |
| van der Wilk et al. (2018) | ✓ | ✓ | ✗ | ✓ | ✓ |
| Schwöbel et al. (2022) | ✓ | ✓ | ✗ | ✓ | ✓ |
| Benton et al. (2020) | ✓ | ✓ | ✓ | ✗ | ✗ |
| **this work** | ✓ | ✓ | ✓ | ✓ | ✓ |

## B  Affine Invariance Parameterisation

We apply the re-parameterization trick (Kingma and Welling, 2013) to define the augmentation distribution as a learnable probability distribution that is differentiable with respect to the parameters:

$$z \in p(\mathbf{x}'|\mathbf{x}, \boldsymbol{\eta}) \rightarrow z = g_{\boldsymbol{\epsilon}}(\mathbf{x}'|\mathbf{x}, \boldsymbol{\eta}), \qquad \boldsymbol{\epsilon} \in U[-1, 1]^k. \tag{17}$$

A general affine parameterization (Benton et al., 2020) can be obtained using $k{=}6$ generator matrices $\mathbf{G}_1, \ldots, \mathbf{G}_6$ and learnable parameters $\boldsymbol{\eta} = [\eta_1, \cdots, \eta_6]^T$ for respective horizontal translation, vertical translations, rotations, horizontal scaling, vertical scaling, and shearing:

$$\mathbf{G}_1 = \begin{bmatrix} 0 & 0 & 1 \\ 0 & 0 & 0 \\ 0 & 0 & 0 \end{bmatrix}, \qquad \mathbf{G}_2 = \begin{bmatrix} 0 & 0 & 0 \\ 0 & 0 & 1 \\ 0 & 0 & 0 \end{bmatrix}, \qquad \mathbf{G}_3 = \begin{bmatrix} 0 & -1 & 0 \\ 1 & 0 & 0 \\ 0 & 0 & 0 \end{bmatrix},$$

$$\mathbf{G}_4 = \begin{bmatrix} 1 & 0 & 0 \\ 0 & 0 & 0 \\ 0 & 0 & 0 \end{bmatrix}, \qquad \mathbf{G}_5 = \begin{bmatrix} 0 & 0 & 0 \\ 0 & 1 & 0 \\ 0 & 0 & 0 \end{bmatrix}, \qquad \mathbf{G}_6 = \begin{bmatrix} 0 & 1 & 0 \\ 1 & 0 & 0 \\ 0 & 0 & 0 \end{bmatrix}.$$

To calculate $\mathbf{x}'_{(x',y')}$, defined to be the value of $\mathbf{x}'$ corresponding location $(x', y') \in \mathbb{R}^2$ on the 2-dimensional grid, we apply an inverse of the forward transformation matrix to find pixel locations in the original image:

$$\begin{bmatrix} x \\ y \end{bmatrix} = \exp\left(\sum_i \epsilon_i \eta_i \mathbf{G}_i\right)^{-1} \begin{bmatrix} x' \\ y' \end{bmatrix} \tag{18}$$

where $\exp(M) = \sum_{n=0}^{\infty} \frac{1}{n!} M^n$ denotes the matrix exponential (Moler and Van Loan, 2003), and the transformations becomes

$$g_{\boldsymbol{\epsilon}}(\mathbf{x}'_{(x',y')}|\mathbf{x}, \boldsymbol{\eta}) = \mathbf{x}_{(x,y)} \tag{19}$$

where pixel values in the output $\mathbf{x}'_{(x',y')}$ are calculated exactly on the grid, and the locations in the input $\mathbf{x}_{(x,y)}$ are obtained through bilinear sampling, which can be, as all of the other steps, automatically differentiated Jaderberg et al. (2015). Furthermore, the entire process is highly efficient as the matrix exponential and inverse are applied on a very small 3x3 matrix and the grid resampling steps are fully parallelizable across all pixels.

# C   Failure cases of Augerino

There are two main failure modes of Augerino that our method overcomes.

**Failure case 1: Augerino requires additional hyperparameter that needs tuning.**    The regularisation used in Augerino introduces an additional hyperparameter that needs tuning. In our experiments, we find that this hyperparameter is non-trivial to tune and requires an additional validation set. Since our method follows from Bayesian model selection, there is no additional hyperparameter. In Fig. 4, we train Augerino on partially-rotated MNIST data using different settings for regulariser hyperparameter $\{0.01, 0.1, 1.0\}$ and compare with our model. We observe that Augerino has difficulty learning partial invariance and is dependent upon the setting of the hyperparameter. Our method, on the other hand, does learn partial rotational invariance without having to tune an additional hyperparameter.

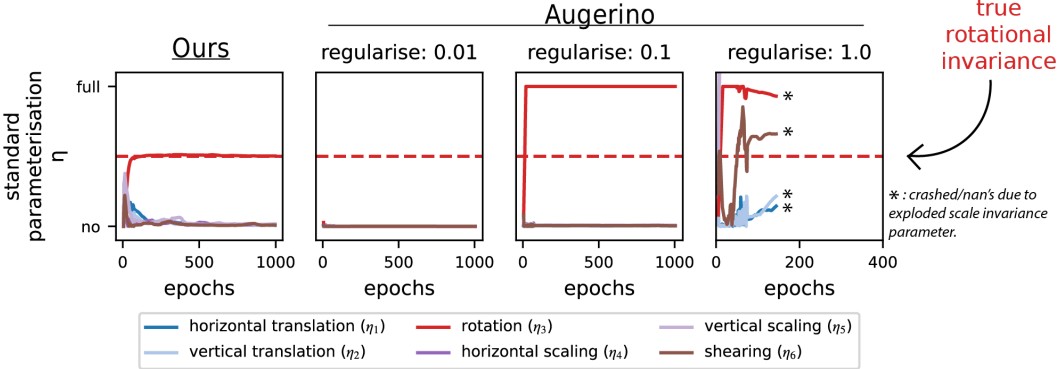

Figure 4: Learning invariance on partially-rotated MNIST with different settings of Augerino regularisation strength.

**Failure case 2: Augerino depends on the used parameterisation of invariance.**    Augerino depends on the parameterisation of the invariance parameter as it uses the heuristic that increasing this parameter corresponds to increased invariance. This makes it fail, for example, under a change of variables $\eta \mapsto \frac{1}{\eta}$. Our method can be applied without requiring knowledge of the particular parameterisation. We demonstrate this failure case in Fig. 5 below, by training our method (left) and Augerino (right) using standard parameterisation $\eta$ (top) and inverted parameterisation $\frac{1}{\eta}$ (bottom). Unlike Augerino, our method learns the correct (in this case rotational) invariance independent of the used parameterisation.

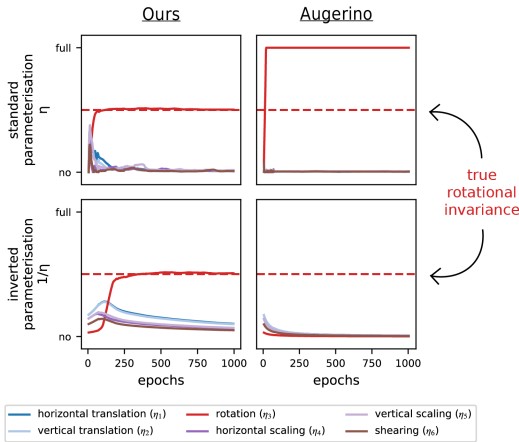

Figure 5: Learned invariance on partially-rotated MNIST for different invariance parameterisations: standard parameterisation $\eta$ (top) and inverted parameterisation $\frac{1}{\eta}$ (bottom). Comparison between our invariance learning approach (left) and Augerino (right). Unlike Augerino, our method is able to learn the correct (in this case rotational) invariance independent of the used parameterisation.

# D   Computational Complexities

In this section, we discuss the computational and memory complexities of the proposed method. In particular, we review the complexity of the GGN approximation, discuss the complexity of the GGN for invariant neural networks, and give the total complexity of *estimating* and *differentiating* the Laplace-GGN approximation using KFAC to the log marginal likelihood, which is necessary for LILA.

## D.1   The complexity of Vanilla GGN Approximations

When not dealing with invariant neural networks that use sampling, the complexities are well-known for diagonal, KFAC, and a full GGN approximation. We have $N$ data points, $C$ neural network outputs, e.g. classes, and $P$ neural network parameters. In general, we assume that $C \ll P$ and also that $C \ll G$, where $G$ is the size of hidden representations.

The full GGN approximation in Eq. 5 is in $\mathcal{O}(NP^2C)$ for computing $N$ matrix-products. Computing its log-determinant additionally costs $\mathcal{O}(P^3)$. The diagonal GGN approximation would be in $\mathcal{O}(NPC)$ and computation of the log-determinant only $\mathcal{O}(P)$. The complexity of KFAC-GGN depends on the type of layer and its in- and output dimensionalities D and G, respectively, as defined in Sec. 4.2. Then, the complexity of computing Eq. 7 is $\mathcal{O}(ND^2 + NCG^2)$. The first term is due to summation of $N$ matrices, each of which is an outer product of $D$-dimensional vectors. The second term is due to the outer product with $\mathbf{\Lambda}_n \in \mathbb{R}^{C \times C}$ in between and since we assume $C \ll G$, we have $CG^2$ at worst for one such product. Computing the log-determinant can be done efficiently in $\mathcal{O}(D^3 + G^3)$ by decomposing the Kronecker factors (Immer et al., 2021a). For typical neural networks, both $D$ and $G$ are below 1000 and computing the log-determinant is tractable.

## D.2   The Effect of Sampling in Invariant Neural Networks on the Complexity

Our invariant formulation of neural networks through perturbations requires $S$ approximation samples. These samples also make computation of the GGN approximations more expensive. The computation of the log determinants remains the same as the sizes of the GGN approximations remain equivalent.

The complexity of a full GGN computation changes to $\mathcal{O}(NP^2C + NPCS)$ where the first term is due to the Jacobian products as above and the second term is for computing and averaging $S$ Jacobians of size $P \times C$ over all $N$ data points. The same term without augmentation samples would be $\mathcal{O}(NPC)$ and is therefore contained in the above complexity $\mathcal{O}(NP^2C)$. Because we usually have $S \ll P$, the complexity of augmented full GGN (Eq. 12) is also unchanged $\mathcal{O}(NP^2C)$.

The complexity of the diagonal-GGN approximation increases relatively worse than the full GGN as it goes from $\mathcal{O}(NPC)$ to $\mathcal{O}(NPCS)$. This is because the diagonal-GGN still requires computation of the $NS$ Jacobians of size $PC$. In total, the computation of the diagonal approximation therefore scales linearly in the number of samples $S$.

The complexity of KFAC-GGN with augmentation samples changes to $\mathcal{O}(N(D^2 + CG^2 + DS + CGS))$ using our proposed method in Eq. 15.[2] The last two terms dependent on $S$ come up due to the aggregation of augmentation samples in our approximation, that is, the expectations over $\mathbf{a}$ and $\mathbf{g}$ in the second line of Eq. 15. However, we typically have that $S \leq D$ and $s \leq G$ and therefore the computational complexity is still in $\mathcal{O}(N(D^2 + CG^2))$. For typical settings, the computational complexity of the KFAC-GGN therefore does not increase over the vanilla variant. This makes the proposed KFAC approximation the only option for invariance learning with differentiable Laplace approximations among the considered ones since the diagonal scales unfavourably in $S$ and the full GGN is too expensive. In principle, one could apply the same approximation as in Eq. 13 for the diagonal GGN but since it even performs worse than the KFAC variant without further approximations (cf. App. F), KFAC is preferred.

If we assume that $D = G$, that is, the in and output size of the layer of interest are equivalent, and we only consider this single fully-connected layer to carry all parameters $P = DG = D^2 = G^2$, it becomes simple to compare the complexities of all three GGN estimators with and without augmentation. This setting is not unrealistic for fully-connected layers where hidden sizes remain rather similar in the middle of the network. The complexities simplify as shown in the table below.

---

[2] If we don't use the proposed additional approximation, the complexity would grow too large to be tractable since the Kronecker-factored structure would not be maintained but lead to a dense Matrix of size $DG \times DG$.

|  | full | KFAC | diagonal |
|---|---|---|---|
| vanilla GGN | $\mathcal{O}(NP^2C)$ | $\mathcal{O}(N(D^2 + CG^2)) = \mathcal{O}(NPC)$ | $\mathcal{O}(NPC)$ |
| augmented GGN | $\mathcal{O}(NP^2C)$ | $\mathcal{O}(N(D^2 + CG^2 + DS + CGS)) = \textcolor{red}{\mathcal{O}(NPC)}$ | $\textcolor{red}{\mathcal{O}(NPCS)}$ |
| (log)det | $\mathcal{O}(P^3)$ | $\mathcal{O}(G^3 + D^3) = \mathcal{O}(P^{1.5})$ | $\mathcal{O}(P)$ |

In this setting the vanilla KFAC-GGN approximation is as cheap to compute as the diagonal approximation and only the computation of the log-determinant is slightly more complex. Even more relevant to us, though, is that the computation of the augmented variant of KFAC has the same computational complexity as the vanilla variant and an $S$ times lower complexity than the diagonal augmented GGN, as marked in red in the table. This is the case although the KFAC variant works better than the diagonal variant (see. App. F). The vanilla KFAC variant often achieves performance on par with the full GGN despite the greatly reduced computational complexity and a much better performance than the diagonal variant (Ritter et al., 2018; Immer et al., 2021a,b; Daxberger et al., 2021).

### D.3 Computational Complexity of Laplace-GGN Estimation and Differentiation

Above, we discussed the computational complexity of the log determinant component of the Laplace-GGN (Eq. 9). The total computational complexity of the Laplace-GGN also depends on the first term, the log likelihood loss, which is independent of the GGN approximation itself with a computational complexity of $\mathcal{O}(NPS)$ and memory complexity $\mathcal{O}(MPS)$ for differentiation since it allows for stochastic batching in $N$. This gives us the total computational complexity for estimating the Laplace-GGN marginal likelihood by summing up the cost of the log likelihood loss ($\mathcal{O}(NPS)$), the cost of the augmented GGN as in the above table, and the cost of computing its log determinant. For our proposed KFAC variant, this gives a total computational complexity of $\mathcal{O}(NPS + NPC + P^{1.5})$ for the simplified architecture assumptions. Naive automatic differentiation would result in the same memory cost for its differentiation, which is intractable. In the next section, we propose a method that allows batching all computation and reducing the memory complexity of differentiating the GGN (or KFAC) to $\mathcal{O}(MPC)$ where $M$ is a free parameter and can be as small as 1 for a single data point, resulting in a total memory complexity of $\mathcal{O}(MPS + MPC)$, where the first term comes from the batch-friendly log likelihood loss. This effectively makes our method possible by reducing the required memory from terabytes to gigabytes in larger scale examples.

### D.4 Measured Computation Time for Different Methods

To give an idea of computation time in practice, we measure parameter and hyperparameter updates on an NVIDIA A30. We report times for training a ResNet (8-8) on CIFAR-10, a CNN on F-MNIST and an MLP on MNIST. We find that parameter updates using maximum likelihood training take 0.16 seconds, 0.08 seconds or 0.06 seconds per epoch, on the respective datasets. When using $S=11$ samples on Augerino and LILA, this becomes 1.9 seconds, 0.5 seconds and 0.3 seconds for the parameter updates in both methods. This is in line with the linear complexity increase of the models in $\mathcal{O}(S)$. The additional hyperparameter update of LILA estimating the marginal likelihood and differentiating it takes 34.5 , 11.1 and 12.9 seconds, for the respective datasets. For the EF approximation, it takes 14.4 , 7.3, and 5.9 seconds, respectively. The improved hyperparameter gradient of LILA increases the cost over Augerino by roughly a factor of 10 in these empirical timings. However, the theoretical analysis above shows that LILA has the same asymptotic complexity as Augerino for $C \le S$ and $\sqrt{P} \le NS$, which commonly holds. We hypothesise that a more efficient implementation could therefore further benefit empirical runtimes.

## E   Details on Efficient Gradient Estimation with GGN and KFAC

In Sec. 4.3, we discussed how to efficiently backpropagate through the log-determinant of the Laplace-GGN objective in Eq. 9. The log-determinant does not allow for batching and straightforward backpropagation because the *memory complexity* of the log-determinant backpropagation is too high as it would be equivalent to the *total computational complexity* of the GGN approximation. In the comparison of complexities (App. D), the total computational complexity would be the sum of computing the GGN approximation and the respective log-determinant computation. Following the example in App. D, using our KFAC-GGN approximation, we would have $\mathcal{O}(NPC + P^{1.5})$ *memory*

*complexity* to compute the gradient with respect to augmentation parameters $\boldsymbol{\eta}$, where the first term is clearly intractable as it is the product of data points $N$ and parameters $P$, even though we are considering a simplified case with $S \ll D = G$ for KFAC and it would normally even scale in the number of samples $S$. The terms other than the log-determinant are simpler to handle, and can be computed using backpropagation: the first term, the conditional log likelihood, can be batched over the $N$ data points due to the sum. The second term, the log prior of parameters $\boldsymbol{\theta}$, is independent of the augmentation parameters.

Our approach relies on computing a *preconditioner* first that incurs a low memory complexity and can be computed in aggregation and then using this result to compute a stochastic, but unbiased, gradient. The preconditioner acts as the vector in a vector-Jacobian product (Paszke et al., 2017). This method allows to decouple computational complexity from memory complexity and is *necessary* to enable learning invariances with the marginal likelihood. For KFAC the computational complexity will be unchanged $\mathcal{O}(NPC + P^{1.5})$ and the memory complexity is $\mathcal{O}(MPC)$ with batch size $M$ as opposed to the intractable $\mathcal{O}(NPC + P^{1.5})$. In Sec. 4.3, we proposed the method in general for full and diagonal GGN log determinants. However, this method does not directly extend to KFAC since it would require evaluation of an expensive Kronecker product. Here, we apply the same idea to show how gradient accumulation is possible for KFAC log determinants.

### E.1 Derivation for KFAC

If we apply the method above directly to KFAC, we would have to compute $\mathbf{H}^{-1}$ for a block corresponding to a single layer which is in most cases too large as it breaks the Kronecker-factored structure. In particular, $\mathbf{H}^{\text{KFAC}} = (\mathbf{A} \otimes \mathbf{G} + \delta\mathbf{I})$ is a typical form that the Kronecker-factored block would take. However, it is inefficient to compute and store such large matrix that is quadratic in the size of the number of parameters of the respective layer. Instead, we would like to maintain the Kronecker-factored structure. Since we need to compute the eigendecomposition of $\mathbf{A}$ and $\mathbf{G}$ to efficiently estimate the marginal likelihood approximation (Immer et al., 2021a), we can make use of it as preconditioner for batched gradient estimation. In fact, it is straightforward to start from eigenvalues $\boldsymbol{\lambda} \in \mathbb{R}^P$ of $\mathbf{H} \in \mathbb{R}^{P \times P}$ instead of $\mathbf{H}$ itself since these can be used to simply compute the log-determinant as a sum of the log of eigenvalues:

$$\log |\mathbf{H}| = \log \prod_{p \in [P]} \lambda_p = \sum_{p \in [P]} \log \lambda_p. \tag{20}$$

For a single block of the Kronecker-factorization and ignoring the normalization by $\frac{1}{N}$ for notational convenience, we have $\mathbf{H} \approx \mathbf{A} \otimes \mathbf{G} + \delta\mathbf{I}$ (Eq. 7) and therefore

$$\log |\mathbf{H}| = \log |\mathbf{A} \otimes \mathbf{G} + \delta\mathbf{I}| = \sum_{a \in [D], g \in [G]} \log(\lambda_a \lambda_g + \delta), \tag{21}$$

where $\lambda_a$ denotes the $a$th eigenvalue of $\mathbf{A}$ and $[D] = 1 \ldots D$ are the dimensions of $\mathbf{A}$ (cf. Sec. 3), and identically for $\lambda_g$. We have a further dependency of $\lambda_a$ and $\lambda_g$ values on the hyperparameter $\eta \in \mathbb{R}$, i.e. $\lambda_a(\eta)$ and $\lambda_g(\eta)$, which is implicit for notational simplicity. Here, we only consider a scalar hyperparameter $\eta$ but the computation, and in particular, the final vector-Jacobian product extend to the vector case. The derivative w.r.t. $\eta$ is given by

$$\frac{\partial}{\partial \eta} \sum_{a,g} \log(\lambda_a \lambda_g + \delta) = \sum_{a,g} (\lambda_a \lambda_g + \delta)^{-1} \frac{\partial}{\partial \eta}(\lambda_a \lambda_g)$$

$$= \sum_{a,g} (\lambda_a \lambda_g + \delta)^{-1} [(\tfrac{\partial}{\partial \eta} \lambda_a) \lambda_g + \lambda_a (\tfrac{\partial}{\partial \eta} \lambda_g)] \tag{22}$$

$$= \sum_a (\tfrac{\partial}{\partial \eta} \lambda_a) \sum_g \lambda_g (\lambda_a \lambda_g + \delta)^{-1} + \sum_g (\tfrac{\partial}{\partial \eta} \lambda_g) \sum_a \lambda_a (\lambda_a \lambda_g + \delta)^{-1}.$$

The two summands in the last expression require the same operations for both $\mathbf{A}$ and $\mathbf{G}$. We will only derive the final expression for the first one, the second one follows accordingly after swapping indices $g$ and $a$. To further simplify, we need the gradient of an eigenvalue $\lambda_a$ with respect to the augmentation. Since $\lambda_a$ is an eigenvalue of $\mathbf{A}$, we will use the chain-rule and have $\frac{\partial}{\partial \eta} \lambda_a = \mathbf{v}_a^\mathsf{T} [\frac{\partial}{\partial \eta} \mathbf{A}] \mathbf{v}_a$ with $\mathbf{v}_a \in \mathbb{R}^D$ as the $a$th eigenvector of $\mathbf{A}$ corresponding the eigenvalue $\lambda_a$.

We can then continue to simplify as follows:

$$\sum_a (\tfrac{\partial}{\partial\eta}\lambda_a) \sum_g \lambda_g (\lambda_a\lambda_g + \delta)^{-1} = \sum_a (\mathbf{v}_a^\mathsf{T}[\tfrac{\partial}{\partial\eta}\mathbf{A}]\mathbf{v}_a) \sum_g \lambda_g (\lambda_a\lambda_g + \delta)^{-1}$$

$$= \sum_{i,j\in[D]} \Big[ \sum_a \mathbf{v}_{a,i}\mathbf{v}_{a,j} \underbrace{\sum_g \lambda_g (\lambda_a\lambda_g + \delta)^{-1}}_{\stackrel{\text{def}}{=} c_a} \Big] \Big[ \tfrac{\partial}{\partial\eta}\mathbf{A}_{i,j} \Big] \quad (23)$$

$$= \mathrm{vec}(\sum_a c_a\mathbf{v}_a\mathbf{v}_a^\mathsf{T})^\mathsf{T} [\textstyle\sum_{n=1}^N \tfrac{\partial}{\partial\eta}\mathrm{vec}(\mathbf{A}_n(\eta))],$$

where the last line can again be expressed as a Jacobian-vector product and the second term sums over the Kronecker factor $\mathbf{A}_n$ per data point $n$ and can be batched or estimated stochastically. In particular, the vector that depends on the eigenvalues $\lambda_.$ and eigenvectors $\mathbf{v}_a$ can be computed after a full dataset pass using a single eigendecomposition of the Kronecker factors. Then, the second term, which requires computing the gradient with respect to the augmentation parameters, can be estimated with low memory footprint by batching and/or a stochastic estimate of the second term.

## F    Discussion of Approximations

We discuss the approximations necessary for the proposed approach to invariance learning with Bayesian model selection. The discussion extends to other complex hyperparameters that could be optimised using the methodology described. Our algorithm is motivated by Bayesian model selection with an empirical Bayes procedure, where we optimise the hyperparameters $\boldsymbol{\eta}$ according to their maximum likelihood on the second level of inference (ML-II). Since the necessary marginal likelihood, $p(\mathcal{D}|\boldsymbol{\eta})$, in Eq. 2 is intractable for neural networks, our approach relies on several approximations detailed below in App. F.1. Despite the approximations, our experiments show that the proposed method is able to recover invariances present in the data without any supervision, which empirically validates the approach. To that end, we show in App. F.2 that the proposed KFAC approximation is necessary to reliably recover invariances while cheaper approximations like the empirical Fisher or diagonal GGN are insufficient.

### F.1    List of Approximations to the Log Marginal Likelihood for an Invariant Model

**Laplace-GGN Approximation.**    Instead of a vanilla Laplace approximation that suffers from estimation issues due to the full-network Hessian as well as potential indefiniteness of it, we use the Laplace-GGN to approximate the log marginal likelihood. The Laplace-GGN (Khan et al., 2019) is more efficient and stable to estimate and has a clear justification due to its equivalence to linearised Laplace (Foong et al., 2019; Immer et al., 2021b; Antorán et al., 2022b). The first step of Laplace-GGN is to linearise a neural network around an arbitrary linearisation point $\boldsymbol{\theta}_*$, i.e., $\mathbf{f}_{\boldsymbol{\theta}_*}^{\text{lin}}(\mathbf{x};\boldsymbol{\theta}) = \mathbf{f}(\mathbf{x};\boldsymbol{\theta}_*) + \mathbf{J}(\mathbf{x};\boldsymbol{\theta}_*)(\boldsymbol{\theta} - \boldsymbol{\theta}_*)$. The error of this Taylor approximation is then in the order of the second derivative of $\mathbf{f}(\mathbf{x};\boldsymbol{\theta})$ at $\boldsymbol{\theta}_*$. Immer et al. (2021b) argue that this changes the model by modifying the underlying likelihood function to depend on $\mathbf{f}^{\text{lin}}$ instead of $\mathbf{f}$ and inference takes place in the first order term $\mathbf{J}(\mathbf{x};\boldsymbol{\theta}_*)\boldsymbol{\theta}$, i.e., we have $p(\mathbf{y}|\mathbf{f}_{\boldsymbol{\theta}_*}^{\text{lin}}(\mathbf{x};\boldsymbol{\theta}),\mathcal{H})$ and an unchanged prior. We can therefore understand such approximation as a modification of the model itself. We arrive at the Laplace-GGN approximation by applying the Laplace approximation to the linearised model. Because we deal with a linear(ised) model, the Laplace approximation tends to perform well with large amounts of data in the case of classification (Bishop, 2006) and is exact for a Gaussian likelihood in the case of regression (Foong et al., 2019). In the large data limit, the Laplace approximation itself becomes asymptotically exact (Dehaene, 2019). Linearisation becomes asymptotically exact in the infinite width limit of neural networks, which results in the neural tangent kernel (Jacot et al., 2018).

**Sampling augmentations.**    The GGN of an invariant model follows from the linearisation of such model (see above paragraph). The only additional approximation arises due to the sampling necessary to approximate the expectation over $p(\mathbf{x}'|\mathbf{x},\boldsymbol{\eta})$ as in Eq. 10. Using Jensen's inequality it can be shown that a Monte Carlo approximation of the invariant neural network in the log likelihood leads

to a lower bound of it. In particular, both Nabarro et al. (2021) and Schwöbel et al. (2022) show

$$\log p(\mathbf{y}|\hat{\mathbf{f}}(\mathbf{x}; \boldsymbol{\theta}, \boldsymbol{\eta})) = \log p(\mathbf{y}|\mathbb{E}_{\mathbf{x}'_1, \dots, \mathbf{x}'_S}[\tfrac{1}{S} \sum_s \mathbf{f}(\mathbf{x}'_s; \boldsymbol{\theta})])$$

$$\geq \mathbb{E}_{\mathbf{x}'_1, \dots, \mathbf{x}'_S}[\log p(\mathbf{y}|\tfrac{1}{S} \sum_s \mathbf{f}(\mathbf{x}'_s; \boldsymbol{\theta}))], \tag{24}$$

where the lower bound is simply due to Jensen's and the fact that we use minimal exponential family likelihoods such that $\mathbf{f}$ is the natural parameter (cf. Sec. 3). In practice, we approximate the last term by sampling one set of augmented inputs $\mathbf{x}'_1, \dots, \mathbf{x}'_S$. The more samples $S$ we take, the tighter the bound. Therefore, it is generally desired to use as many samples as affordable.

**Augmented KFAC approximation.**   KFAC of an invariant neural network relies on further approximations to the intractable GGN matrix that is quadratic in the number of neural network parameters. We first discuss the approximations of vanilla KFAC before the additional approximation that is necessary for invariant neural networks. KFAC approximates the full $P \times P$ GGN by a block-diagonal approximation that captures the GGN over each layer independently in form of a block-diagonal matrix. Further, to maintain an efficient Kronecker-factored structure that is only exact for a single data point (cf. Eq. 6) a product of sums is approximated by sums of products (cf. Eq. 7). In practice, this approximation has been validated in the context of optimisation (Martens and Grosse, 2015; Osawa et al., 2020; Dangel et al., 2019), posterior approximation (Ritter et al., 2018; Zhang et al., 2018; Osawa et al., 2019; Immer et al., 2021b), and model selection (Immer et al., 2021a, 2022). Both approximations are only exact when the model is linear. The additional approximation proposed by us applies to the augmentation samples and is similar to the second approximation of vanilla KFAC but applied to the Jacobians of the invariant neural network in Eq. 13. In our experiments, we empirically verify that such approximation does not lead to issues. Below in App. F.2, we show that this approximation still leads to better performance than a diagonal GGN that avoids it. For an invariant linear model, the proposed KFAC extension would be exact as was the case for vanilla KFAC:

$$\hat{\mathbf{J}}_{\boldsymbol{\theta}}(\mathbf{x}_n; \boldsymbol{\eta}) = \mathbb{E}_{p(\mathbf{x}'_n|\mathbf{x}_n, \boldsymbol{\eta})}[\mathbf{x}'_n \otimes \mathbf{I}] = \mathbb{E}_{p(\mathbf{x}'_n|\mathbf{x}_n, \boldsymbol{\eta})}[\mathbf{x}'_n] \otimes \mathbf{I} \tag{25}$$

gives the Jacobian of an invariant linear model and requires no further approximation since the product separates naturally. This is also the case for the last layer of a neural network.

### F.2   Diagonal and Empirical Fisher Approximations

Here, we briefly discuss the reason for extending the KFAC-GGN approximation to invariant neural networks instead of using a diagonal approximation. The argument is both computational, following from App. D, and empirical, following from the better performance of KFAC than diagonal. We further discuss the empirical Fisher, which is an alternative to the GGN but has a $C$ times lower computational complexity where $C$ is the number of outputs (classes). The empirical Fisher is a viable alternative but is not justifiable by a linearisation like the GGN and is known to have certain pathologies (Kunstner et al., 2019). Fig. 6 shows the invariance learning performances for partially rotated ($\pm 90°$) FashionMNIST with KFAC and diagonal each with GGN and empirical Fisher variant. The setup is identical to the one used in the experiments and is conducted on 5000 randomly chosen data points over 3 seeds. The figure only contains the run of a single seed but the observation is consistent across all runs.

**Diagonal approximations.**   Although it does not model any correlations, a diagonal GGN approximation is typically only marginally more efficient than a KFAC approximation (cf. App. D for a detailed discussion of complexities). This is due to the efficient approximations employed in KFAC that could potentially lead to a worse overall performance but empirically it has been found that KFAC performs always at least as well as a diagonal approximation as discussed in App. F.1. When considering invariance learning with GGN approximations, the proposed KFAC for invariant neural networks further significantly reduces the runtime and memory complexity in the augmentation samples $S$, which we discuss in App. D. Fig. 6 further shows that KFAC performs significantly better in terms of test accuracy and in terms of recovering the underlying $\pm 90°$ rotational invariance. Overall, this leaves no reason to use a diagonal approximation for learning complex hyperparameters.

**Empirical Fisher.**   The empirical Fisher is an approximation to the true Fisher, which is in turn equivalent to the GGN for the cases we consider, but can be significantly cheaper to compute. For example in the case of classification with $C$ classes, the cost of computing the GGN scales linearly in

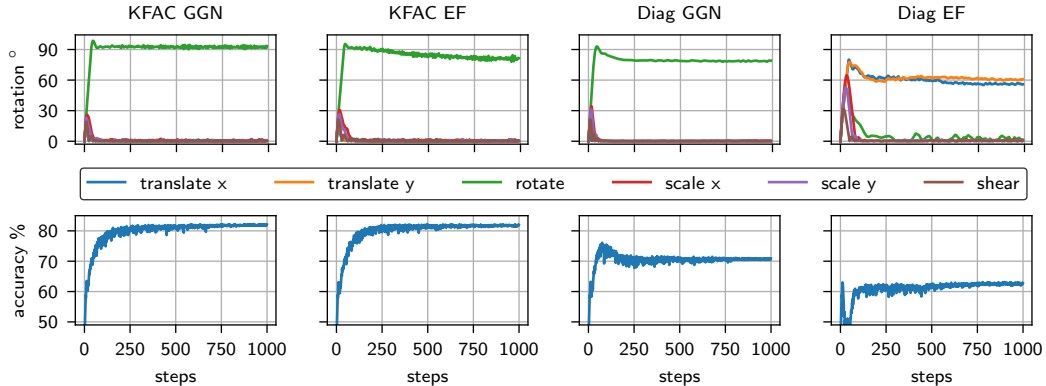

Figure 6: Comparison of KFAC and diagonal GGN and EF approximations for invariance learning using the corresponding log marginal likelihood approximation on 5000 random ±90° rotated FashionMNIST samples. KFAC performs significantly better than diagonal approximations. GGN is slightly better than EF for KFAC but significantly better in the diagonal case.

$C$ while the empirical Fisher does not. Mathematically, the empirical Fisher is given by summing up outer products of gradients as opposed to Jacobians. However, the empirical Fisher does not follow from a linearisation perspective and also has certain pathologies in optimisation (Kunstner et al., 2019). In Fig. 6, we find that the empirical Fisher can work as well as the GGN for KFAC but its diagonal approximation fails and even learns the wrong invariances. That the diagonal empirical Fisher, the cheapest curvature approximation, fails in this setting is interesting because Immer et al. (2021a) observe that it can reliably learn regularisation hyperparameters for ResNets. We hypothesise that learning invariances requires better approximations and profits from approximation quality. The KFAC empirical Fisher can be extended to invariance learning similar to the GGN and requires averaging the gradients, which are used for the outer products, over augmentation samples. In Table 2, we additionally compare its performance for a wide ResNet on standard CIFAR-10, where it performs almost as well but with 10-fold speed-up.

## G  Mechanism of Differentiable Laplace for Invariance Learning

We discuss the mechanism by which the Laplace approximation to the marginal likelihood enables invariance learning and why simple maximum likelihood is insufficient. Leaving out terms that do not depend on the invariance parameter $\boldsymbol{\eta}$, such as the prior $\log p(\boldsymbol{\theta}_*)$, the Laplace-GGN approximation to the log marginal likelihood of an invariant neural network introduced in Eq. 9 can be decomposed as

$$\underbrace{\underbrace{\sum_{n=1}^{N} \log p(\mathbf{y}_n | \frac{1}{S}\sum_s \mathbf{f}(\mathbf{g}(\mathbf{x}_n, \boldsymbol{\epsilon}_s; \boldsymbol{\eta}); \boldsymbol{\theta}_*))}_{\text{maximum likelihood objective}} - \frac{1}{2}\log |\hat{\mathbf{H}}_{\boldsymbol{\theta}_*}^{\text{GGN}}(\boldsymbol{\eta})|}_{\text{marginal likelihood objective}} \ . \tag{26}$$

In the following we discuss why the maximum likelihood objective for $\boldsymbol{\eta}$ in general will not lead to proper invariance learning while the marginal likelihood objective will.

### G.1  Regular Maximum Likelihood Does *Not* Learn Invariances

In the context of deep learning, the underlying non-invariant neural network function $\mathbf{f}$ is complex and can arbitrarily (over-)fit the data. Assuming such complex function, we do not need any invariance parameter to fit the data optimally according to the maximum likelihood part in above Eq. 26. In fact, changing the invariance parameter $\boldsymbol{\eta}$ rather hinders fitting due to sampling noise and restricting the function class. This behaviour is shown on fully rotated MNIST in the figure on the right where all affine invariance parameters, also rotation, remain zero.

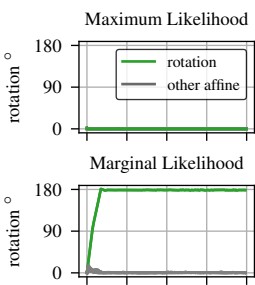

## G.2 Marginal Likelihood Learns Invariances

The marginal likelihood objective arises from the principle of Bayesian model selection, which trades off performance and model complexity and is introduced in common machine learning literature (MacKay, 2003, Sec. 28; Bishop, 2006, Sec. 3.4 and 4.4; Murphy, 2012, Sec. 5.6) and briefly in Sec. 3. *Intuitively*, a model is less complex if it can explain two data points $\mathbf{x}, \mathbf{x}'$, where one is a transformed version of the other, with the same function. A non-invariant model needs to fit both data points individually and is therefore more complex. Van der Wilk et al. (2018) first introduced invariance learning with the marginal likelihood for Gaussian processes and discuss the underlying principle of it. In the following, we elaborate on the mechanism when using the Laplace-GGN approximation to the log marginal likelihood for invariance learning proposed in this work.

*Mathematically*, the log-determinant part of the Laplace-GGN marginal likelihood approximation, which is the only additional term in comparison to the maximum likelihood objective, favours invariant models. Consider two (or more) data points $\mathbf{x}_1$ and $\mathbf{x}_2$ in the same orbit, i.e. one is a transformed version of the other, and we know the true invariance parameter $\boldsymbol{\eta}_*$. Further, we assume to have exact invariance in the data such that the perturbation distribution is uniform on the orbit of the corresponding group (Sec. 3; Kondor et al., 2018; Ginsbourger et al., 2016). This means that $\mathbf{x}_1$ is an augmented version of $\mathbf{x}_2$ and vice versa and their perturbation distributions are identical: $\forall \mathbf{x}' : p(\mathbf{x}'|\mathbf{x}_1, \boldsymbol{\eta}_*) = p(\mathbf{x}'|\mathbf{x}_2, \boldsymbol{\eta}_*)$. Then, an invariant model would have identical Jacobians for both data points, which leads to zero angle between Jacobians and maximizes the negative log determinant. To see this, we use a reformulation using the matrix determinant Lemma as proposed by Immer et al. (2021a) that allows analytic computation for two data points. Defining $\hat{\mathbf{J}} \in \mathbb{R}^{2C \times P}$ that consists of the two Jacobians $\mathbf{J}_i = \hat{\mathbf{J}}_{\boldsymbol{\theta}_*}(\mathbf{x}_i; \boldsymbol{\eta}_*) \in \mathbb{R}^{C \times P}$ for $i \in \{1, 2\}$ and $\hat{\boldsymbol{\Lambda}} \in \mathbb{R}^{2C \times 2C}$ that is a block-diagonal matrix with entries $\hat{\boldsymbol{\Lambda}}(\mathbf{x}_i; \boldsymbol{\theta}_*, \boldsymbol{\eta}_*) \in \mathbb{R}^{C \times C}$ for $i \in \{1, 2\}$, we have

$$-\log|\hat{\mathbf{H}}_{\boldsymbol{\theta}_*}^{\text{GGN}}(\boldsymbol{\eta}_*)| = -\log|\hat{\mathbf{J}}^\mathsf{T}\hat{\boldsymbol{\Lambda}}\hat{\mathbf{J}} + \delta\mathbf{I}_P| \propto -\log\left[|\delta^{-1}\hat{\mathbf{J}}\hat{\mathbf{J}}^\mathsf{T} + \hat{\boldsymbol{\Lambda}}^{-1}||\hat{\boldsymbol{\Lambda}}|\right], \qquad (27)$$

where $\delta$ denotes the prior precision of a Gaussian prior. For simplicity, assume a Gaussian likelihood with observation noise $\sigma = 1$, prior $\delta = 1$, and a single output $C = 1$. Then, we have

$$-\log|\hat{\mathbf{H}}_{\boldsymbol{\theta}_*}^{\text{GGN}}(\boldsymbol{\eta}_*)| \propto -\log|\hat{\mathbf{J}}\hat{\mathbf{J}}^\mathsf{T} + \mathbf{I}_2| = -\log\left(1 + \|\mathbf{J}_1\|_2^2 + \|\mathbf{J}_2\|_2^2 + \|\mathbf{J}_1\|_2^2\|\mathbf{J}_2\|_2^2 - (\mathbf{J}_1\mathbf{J}_2^\mathsf{T})^2\right)$$
$$= -\log\left(1 + \|\mathbf{J}_1\|_2^2 + \|\mathbf{J}_2\|_2^2 + (1 - \cos^2\phi)\|\mathbf{J}_1\|_2^2\|\mathbf{J}_2\|_2^2\right), \qquad (28)$$

where $\phi$ is the angle between Jacobians $\mathbf{J}_1, \mathbf{J}_2$ of $\mathbf{x}_1$ and $\mathbf{x}_2$, respectively. Therefore assuming fixed Jacobian norms, the negative log determinant is maximized when we have $\cos^2\phi = 1$, i.e., when Jacobians have zero angle $\phi = 0$ modulo direction.

# H    Detailed Algorithm

Here, we detail the final algorithm that we proposed for invariance learning using marginal likelihood approximations. The algorithm extends the one proposed by Immer et al. (2021a) to complex hyperparameters. Their algorithm is only tractable for simple hyperparameters, such as regularisation or observation noise, here denoted by $\mathcal{M}$. Our contributions enable scaling to invariance parameters and other complex hyperparameters and are detailed in Lines 14 to 21 in Alg. 1. The algorithm below uses simple stochastic gradient updates with a fixed learning rate. In practise, we use a decaying step size and the Adam optimiser (Kingma and Ba, 2015) as detailed in the training procedure for our experiments in App. I.

**Algorithm 1** Detailed invariance learning algorithm using marginal likelihood estimates.
Lines 4-13 are as in (Immer et al., 2021a), 14-21 is our extension to invariance parameters.

---

1: **Input:** dataset $\mathcal{D} = \{(\mathbf{x}_n, \mathbf{y}_n)\}_{n=1}^N$, simple hyperparams (observation and prior noise) $\mathcal{M}$, invariance parameter $\boldsymbol{\eta}$, likelihood $p(\mathcal{D}|\boldsymbol{\theta}, \boldsymbol{\eta}, \mathcal{M})$, prior $p(\boldsymbol{\theta}|\mathcal{M})$, number of MC samples $S$, batch size $M$, number of epochs $T$, neural network learning rate $\gamma_{\boldsymbol{\theta}}$, hyperparamater learning rate $\gamma_{\mathcal{M}}$, invariance parameter learning rate $\gamma_{\boldsymbol{\eta}}$

2: **Initialise:** neural network parameters $\boldsymbol{\theta}$ (e.g. He et al. (2016) init.), $\mathcal{M}$ and $\boldsymbol{\eta} = \mathbf{0}$

3: **for** $T$ epochs **do**

4:     **for** each batch $\mathcal{B} \subseteq \mathcal{D}$ of size $|\mathcal{B}| \leq M$ **do**         ▷ SGD training of neural network

5:         $\boldsymbol{\theta} \leftarrow \boldsymbol{\theta} + \gamma_{\boldsymbol{\theta}} \nabla_{\boldsymbol{\theta}} \frac{N}{|\mathcal{B}|} \sum_{m \in \mathcal{B}} \log p(\mathbf{y}_m | \frac{1}{S} \sum_{s=1}^S \mathbf{f}(\mathbf{g}(\mathbf{x}_n, \boldsymbol{\epsilon}_s; \boldsymbol{\eta}); \boldsymbol{\theta}), \mathcal{M}) + \log p(\boldsymbol{\theta}|\mathcal{M})$

6:     **end for**

7:     $\hat{\mathbf{H}}_{\boldsymbol{\theta}}^{\text{GGN}} \leftarrow \mathbf{0}$

8:     **for** each batch $\mathcal{B} \subseteq \mathcal{D}$ of size $|\mathcal{B}| \leq M$ **do**         ▷ Compute $\hat{\mathbf{f}}$ and GGN without comp. graph

9:         $\hat{\mathbf{f}}_{m \in \mathcal{B}} \leftarrow [\frac{1}{S} \sum_{s=1}^S \mathbf{f}(\mathbf{g}(\mathbf{x}_m, \boldsymbol{\epsilon}_s; \boldsymbol{\eta}); \boldsymbol{\theta})]_{m \in \mathcal{B}}$

10:         $\hat{\mathbf{H}}_{\boldsymbol{\theta}}^{\text{GGN}} \leftarrow \hat{\mathbf{H}}_{\boldsymbol{\theta}}^{\text{GGN}} + \sum_{m \in \mathcal{B}} \hat{\mathbf{H}}_m^{\text{GGN}}$         ▷ Typically use KFAC

11:     **end for**

12:     $\hat{\mathbf{H}}_{\boldsymbol{\theta}}(\mathcal{M}) \leftarrow \alpha_{\mathcal{M}} \hat{\mathbf{H}}_{\boldsymbol{\theta}}^{\text{GGN}} - \nabla_{\boldsymbol{\theta}}^2 \log p(\boldsymbol{\theta}|\mathcal{M})$         ▷ $\alpha_{\mathcal{M}}$ can model observation noise

13:     $\mathcal{M} \leftarrow \mathcal{M} + \gamma_{\mathcal{M}} \nabla_{\mathcal{M}} \sum_{n=1}^N \log p(\mathbf{y}_n | \hat{\mathbf{f}}_n, \mathcal{M}) + \log p(\boldsymbol{\theta}|\mathcal{M}) - \frac{1}{2} \log \left| \frac{1}{2\pi} \hat{\mathbf{H}}_{\boldsymbol{\theta}}(\mathcal{M}) \right|$

14:     $\hat{\mathbf{h}} \leftarrow \text{vec}(\hat{\mathbf{H}}_{\boldsymbol{\theta}}^{-1})$         ▷ Compute preconditioner of vjp; for KFAC see App. E.1

15:     $\mathbf{g}_{\boldsymbol{\eta}} \leftarrow \mathbf{0}$

16:     **for** each batch $\mathcal{B} \subseteq \mathcal{D}$ of size $|\mathcal{B}| \leq M$ **do**         ▷ Aggregate invariance gradient

17:         $\mathbf{g}_{\boldsymbol{\eta}} \leftarrow \mathbf{g}_{\boldsymbol{\eta}} + \nabla_{\boldsymbol{\eta}} \sum_{m \in \mathcal{B}} \log p(\mathbf{y}_m | \mathbf{f}(\mathbf{g}(\mathbf{x}_m, \boldsymbol{\epsilon}_s; \boldsymbol{\eta}); \boldsymbol{\theta}))$

18:         $\hat{\mathbf{h}}_{\mathcal{B}}(\boldsymbol{\eta}) \leftarrow \text{vec}(\sum_{m \in \mathcal{B}} \hat{\mathbf{H}}_m^{\text{GGN}}(\boldsymbol{\eta}))$         ▷ Compute with graph to $\boldsymbol{\eta}$

19:         $\mathbf{g}_{\boldsymbol{\eta}} \leftarrow \mathbf{g}_{\boldsymbol{\eta}} + \text{vjp}(\hat{\mathbf{h}}, \hat{\mathbf{h}}_{\mathcal{B}}(\boldsymbol{\eta}), \boldsymbol{\eta})$         ▷ vector-Jacobian product w.r.t $\boldsymbol{\eta}$

20:     **end for**

21:     $\boldsymbol{\eta} \leftarrow \boldsymbol{\eta} + \gamma_{\boldsymbol{\eta}} \mathbf{g}_{\boldsymbol{\eta}}$

22: **end for**

23: **Return:** optimised neural network, invariance, and hyperparameters $(\boldsymbol{\theta}, \boldsymbol{\eta}, \mathcal{M})$, approximation to log marginal likelihood $\log p(\mathcal{D}|\boldsymbol{\eta}, \mathcal{M})$, and optionally posterior approximation $p(\boldsymbol{\theta}|\mathcal{D}, \boldsymbol{\eta}, \mathcal{M})$.

---

# I Training Details

The code for LILA and the experiments is available at `https://github.com/tychovdo/lila`.

## I.1 Dataset Details

We used MNIST LeCun and Cortes (2010), FashionMNIST Xiao et al. (2017) and CIFAR-10 Krizhevsky et al. (2009) in our experiments. MNIST and FashionMNIST pixel values are scaled to the interval $[0, 1]$ and CIFAR-10 images are standardised to zero mean and unit variance per channel following common practise. We created the following transformed datasets to validate the invariance learning of our method:

- Partially rotated dataset: Each sample rotated with randomly sampled radian angle from $U[-\frac{\pi}{2}, \frac{\pi}{2}]$.
- Fully rotated dataset: Each sample rotated with randomly sampled radian angle from $U[-\pi, \pi]$.
- Translated dataset: Each sample is translated by $dx$ pixels in x-direction and $dy$ pixels in the y-direction, both independently sampled from $dx, dy \sim U[-8, 8]$.
- Scaled dataset: Each sample scaled around center with $\exp(s)$ pixels in both x-direction and y-direction simultaneously, sampled from $s \sim U[-\log(2), \log(2)]$.

## I.2 Network Architectures

For the MLP we use a single hidden layer with 1000 hidden units and a `tanh` activation function. For the CNN experiments we used a convolutional neural network with three convolutional layers with $3 \times 3$ filters, 1 stride, 1 padding, bias weights with increasing channels sizes (3-16-32-64) followed by a linear layer with 256 hidden units with `ReLU` activation function between layers.

For CIFAR-10, we used ResNets with fixup parameterization and initialization (Zhang et al., 2019) to avoid batch norm, which conflicts with a Bayesian model (Wenzel et al., 2020). The ResNets sizes are indicated by (input-channels - width) in the tables and figures. We use ResNets (8-8) and (8-16) except in Table 2, where we use a wide ResNet (Zagoruyko and Komodakis, 2016) 16-4 with fixup (Zhang et al., 2019) as in (Daxberger et al., 2021). The WRN 16-4 is only run on plain CIFAR-10 as they are expensive to run for all subset sizes and transformations. All vanilla ResNets can fit the training data to $100\%$ accuracy. Following Immer et al. (2021a), we use prior precision hyperparameters $\mathcal{M}$ per neural network layer and use a learning rate of $0.05$ for hyperparameters.

### I.3 Training Parameters

For the MNIST and FashionMNIST experiments, we trained our models for 1000 epochs with a batch size of 1000 and 31 augmentation samples. The Adam optimizer was used with a learning rate of 0.005 cosine decayed to 0.0001 and a momentum of 0.9 for the network weights, and for the invariance and hyperparameters we used a learning rate of 0.05 together with a 10 epochs burn-in period. Each experiment was repeated 3 times with different random seeds. For the MNIST and FashionMNIST subset of data results in App. J.5, we use the same hyperparameters and 1000 epochs for subset sizes $[312, 1250, 5000, 20000]$.

For the CIFAR-10 experiments we trained the ResNets for 200 epochs on the full data with initial learning rate of $0.1$ using SGD with momentum of $0.9$ and cosine learning rate decay to $10^{-6}$. We use a batch size of $250$ and accumulate gradients with respect to augmentation parameters for $M = 5000$ data points using $S = 20$ augmentation samples. We optimise $\boldsymbol{\eta}$ using Adam (Kingma and Ba, 2015) starting from epoch 10 with learning rate 0.005. For Augerino (Benton et al., 2020) we use the same learning rate, 0.005 for $\boldsymbol{\eta}$ and default weight-decay of $10^{-4}$ as used in their experiments. For the CIFAR-10 subset of data results in App. J.5, we use the same hyperparameters but $[600, 400, 300, 250]$ epochs for subset sizes $[1000, 5000, 10000, 20000]$, respectively. We report the mean and one standard error over three seeds for all experiments.

### I.4 Classification Example

For the toy classification example with $N = 200$ data points, we use a fully-connected neural network with a single hidden layer of $50$ neurons and `tanh` activation. All methods train with $500$ steps with a learning rate of $0.1$ using the Adam optimiser (Kingma and Ba, 2015). For the prior precision hyperparameter in $\mathcal{M}$, we use the same learning rate of $0.1$. For the augmentation parameter $\boldsymbol{\eta}$, we use a lower learning rate of $0.005$ and decay it with cosine-decay to $0.0001$ due to the stochasticity of the gradients. For data augmentation and our method, we use $S = 100$ perturbation samples. Each step, we optimise neural network parameters, invariance parameters $\boldsymbol{\eta}$, and scale hyperparameters $\mathcal{M}$, which is a single prior precision in this case. We plot the improved posterior predictive for Laplace approximations (Immer et al., 2021b) in both Fig. 1 and Fig. 7. The log marginal likelihood values given in Fig. 7 are all computed with a full Laplace-GGN, even for the KFAC variant, for comparability of the estimates. The hyperparameters $\mathcal{M}, \boldsymbol{\eta}$ in Fig. 7 (e) are optimised using KFAC for invariant neural networks proposed in Sec. 4.

## J  Additional Results and Experiments

### J.1  Classification Example

In Fig. 7, we compare our approach using the standard GGN and the KFAC-GGN with three baselines on a classification example that was generated with a soft $\pm 60°$ rotational invariance around the origin. We consider only rotational invariance about the origin with parameter $\eta_{\text{rot}}$. Our method with both full and KFAC GGN successfully learns the rotational invariance and obtains a similar value for $\eta_{\text{rot}}$ as the data generating process and obtains the best marginal likelihood indicating a better generalisation. The non-invariant model achieves a worse marginal likelihood and standard data augmentation leads to an even lower value due to problems with its likelihood (Nabarro et al., 2021). A model in polar coordinates corresponds to incorporating prior knowledge manually and, as expected, performs on par with our learned model. Our model, which can explicitly learn invariances, attains a slightly better log marginal likelihood since the polar model is not restricted to be fully invariant as the prediction can vary along the rotational angle.

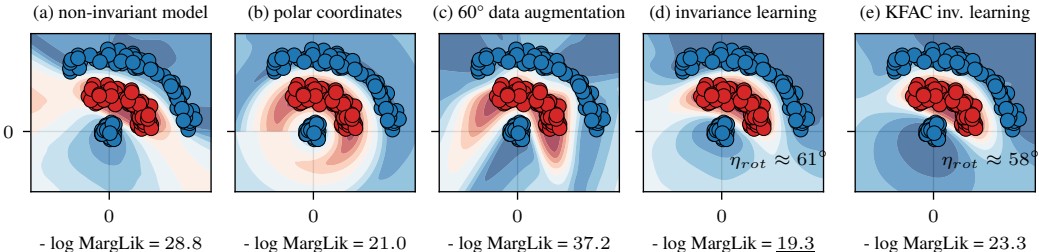

| (a) non-invariant model | (b) polar coordinates | (c) 60° data augmentation | (d) invariance learning | (e) KFAC inv. learning |
|---|---|---|---|---|
| - log MargLik = 28.8 | - log MargLik = 21.0 | - log MargLik = 37.2 | - log MargLik = 19.3 | - log MargLik = 23.3 |

Figure 7: Different approaches to (approximately) invariant neural networks compared to a non-invariant one (a) and their corresponding Laplace log marginal likelihoods. (b): data representation in polar coordinates leads to a good marginal likelihood but requires prior knowledge. (c): standard data augmentation conflicts with the Bayesian model and achieves worse marginal likelihood. (d): treating the invariance or data augmentation parameter as part of a Bayesian model and optimising it allows to obtain the best marginal likelihood. (e): our efficient approximation based on an augmented KFAC Laplace approximation performs almost as well as the full Gauss-Newton in (d).

## J.2 Quantitative Results

Table 4: Marginal Likelihood and Test Accuracy for models using GGN-KFAC on different versions of the MNIST, FashionMNIST and CIFAR-10 datasets. Our proposed method outperforms the non-invariants for almost all models and datasets, in terms of both measures.

| Network | Dataset | Model | Marginal Likelihood | | | | | Test Accuracy | | | | |
| | | | Fully rotated Dataset | Partially rotated Dataset | Translated Dataset | Scaled Dataset | Original Dataset | Fully rotated Dataset | Partially rotated Dataset | Translated Dataset | Scaled Dataset | Original Dataset |
|---|---|---|---|---|---|---|---|---|---|---|---|---|
| MNIST | MLP | non-invariant | -31.3k ±99 | -23.1k ±29 | -36.0k ±10 | -17.5k ±28 | -10.5k ±5 | 93.82 ±0.10 | 95.83 ±0.03 | 94.15 ±0.02 | 97.07 ±0.06 | 98.20 ±0.03 |
| | | Augerino | - | - | - | - | - | **97.83** ±0.03 | 96.35 ±0.02 | 94.47 ±0.08 | 97.45 ±0.03 | 98.45 ±0.03 |
| | | LILA (**ours**) | **-9.7k** ±25 | **-10.5k** ±27 | **-14.5k** ±11 | **-11.3k** ±22 | **-6.2k** ±33 | 97.74 ±0.07 | **97.81** ±0.11 | **97.28** ±0.05 | **98.33** ±0.05 | **98.98** ±0.05 |
| | CNN | non-invariant | -14.6k ±1626 | -10.7k ±1070 | -16.5k ±1014 | -9.5k ±225 | -5.0k ±428 | 95.97 ±0.33 | 97.51 ±0.17 | 96.54 ±0.29 | 98.37 ±0.00 | 99.09 ±0.02 |
| | | Augerino | - | - | - | - | - | **99.04** ±0.02 | 98.91 ±0.03 | 97.79 ±0.09 | 98.77 ±0.06 | 98.26 ±0.10 |
| | | LILA (**ours**) | **-6.9k** ±77 | **-7.2k** ±295 | **-8.6k** ±136 | **-7.5k** ±54 | **-4.3k** ±27 | 98.83 ±0.07 | **98.92** ±0.05 | **98.69** ±0.07 | **99.01** ±0.06 | **99.42** ±0.02 |
| F-MNIST | MLP | non-invariant | -55.1k ±117 | -46.2k ±184 | -52.4k ±36 | -39.2k ±27 | -25.5k ±6 | 77.62 ±0.30 | 81.10 ±0.23 | 77.68 ±0.10 | 81.84 ±0.05 | 88.48 ±0.56 |
| | | Augerino | - | - | - | - | - | 77.76 ±0.15 | 81.40 ±0.05 | 78.05 ±0.10 | 82.46 ±0.09 | 89.10 ±0.13 |
| | | LILA (**ours**) | **-29.3k** ±63 | **-29.4k** ±148 | **-36.6k** ±17 | **-34.7k** ±201 | **-22.6k** ±42 | **87.39** ±0.06 | 86.72 ±0.13 | **84.62** ±0.08 | **84.31** ±0.06 | **89.94** ±0.12 |
| | CNN | non-invariant | -58.7k ±189 | -49.9k ±275 | -53.8k ±257 | -43.3k ±488 | -28.9k ±165 | 78.69 ±0.28 | 82.12 ±0.35 | 80.33 ±0.19 | 83.66 ±0.37 | 89.54 ±0.23 |
| | | Augerino | - | - | - | - | - | 85.76 ±3.23 | 81.54 ±0.19 | 82.94 ±0.13 | 83.58 ±0.08 | 90.07 ±0.12 |
| | | LILA (**ours**) | **-26.5k** ±101 | **-27.2k** ±0 | **-29.2k** ±65 | **-30.1k** ±0 | **-21.0k** ±233 | **89.45** ±0.03 | **88.40** ±0.00 | **87.73** ±0.20 | **87.33** ±0.00 | **91.92** ±0.21 |
| CIFAR-10 | ResNet (8-8) | non-invariant | -73.1k ±152 | -67.5k ±334 | -53.7k ±154 | -59.3k ±749 | -46.5k ±345 | 51.14 ±0.47 | 55.29 ±0.70 | 64.84 ±0.21 | 59.81 ±0.96 | 69.74 ±0.75 |
| | | Augerino | - | - | - | - | - | 70.88 ±0.17 | 70.95 ±0.26 | **74.44** ±0.24 | 70.67 ±0.28 | 79.34 ±0.36 |
| | | LILA (**ours**) | **-51.1k** ±244 | **-47.8k** ±296 | **-38.6k** ±181 | **-41.7k** ±225 | **-31.2k** ±346 | **71.06** ±0.47 | **73.03** ±0.45 | 74.18 ±0.07 | **71.54** ±0.18 | **80.22** ±0.24 |
| | ResNet (8-16) | non-invariant | -80.9k ±483 | -72.8k ±133 | -56.2k ±329 | -62.0k ±283 | -46.1k ±953 | 54.16 ±0.40 | 59.90 ±0.12 | 69.65 ±0.16 | 66.06 ±0.13 | 74.13 ±0.51 |
| | | Augerino | - | - | - | - | - | 75.40 ±0.19 | 74.76 ±0.34 | 73.71 ±0.31 | 72.07 ±0.09 | 79.03 ±1.04 |
| | | LILA (**ours**) | **-43.1k** ±217 | **-38.7k** ±135 | **-35.4k** ±1969 | **-40.2k** ±3404 | **-30.2k** ±2344 | **79.50** ±0.62 | **77.71** ±0.46 | **79.21** ±0.17 | **76.03** ±0.15 | **84.19** ±0.76 |

## J.3 Additional Results: Invariance Learning Trajectories on All Datasets

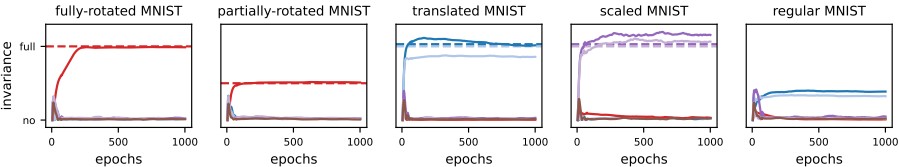

Figure 8: MLP on MNIST

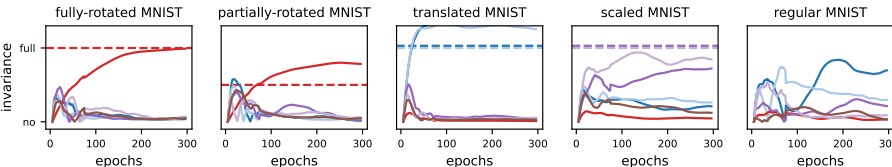

Figure 9: CNN on MNIST

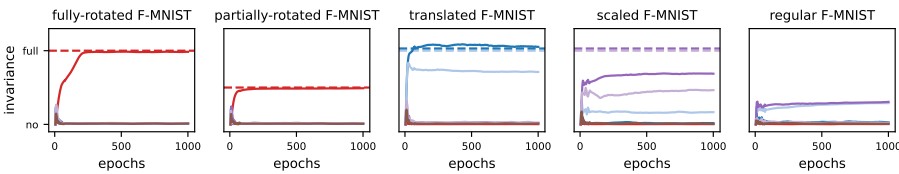

Figure 10: MLP on F-MNIST

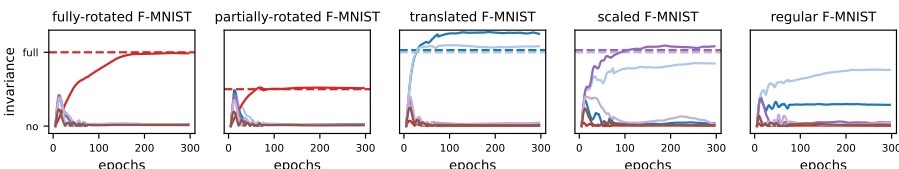

Figure 11: CNN on F-MNIST

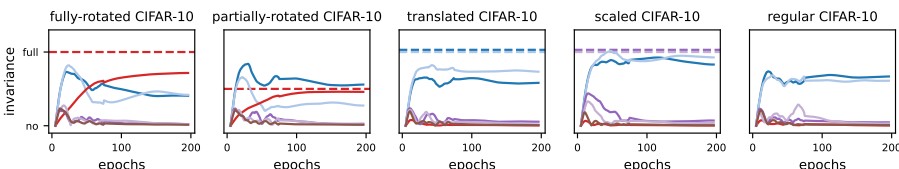

Figure 12: ResNet on CIFAR-10

## J.4 Additional Results: Learned Invariance Barplots and Samples on all Datasets

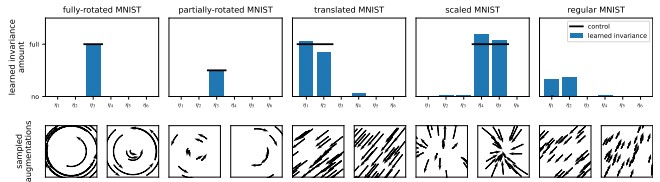

Figure 13: MLP on MNIST

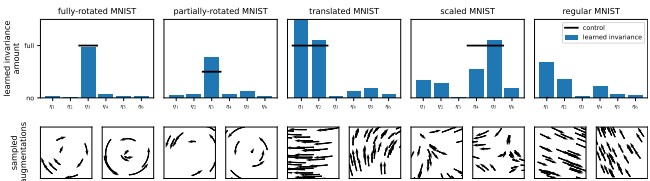

Figure 14: CNN on MNIST

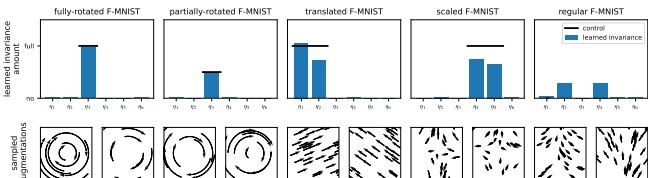

Figure 15: MLP on F-MNIST

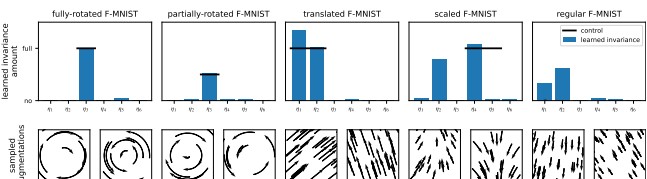

Figure 16: CNN on F-MNIST

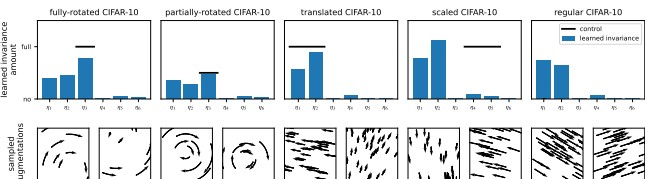

Figure 17: ResNet on CIFAR-10

## J.5 Additional Results: Subset Experiments on All Datasets

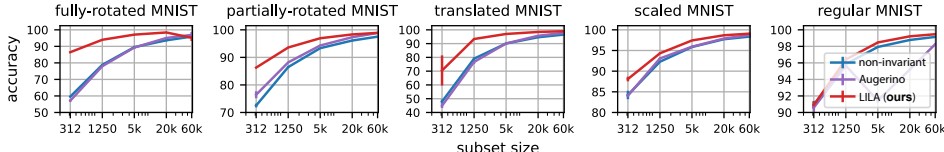

Figure 18: MLP on MNIST

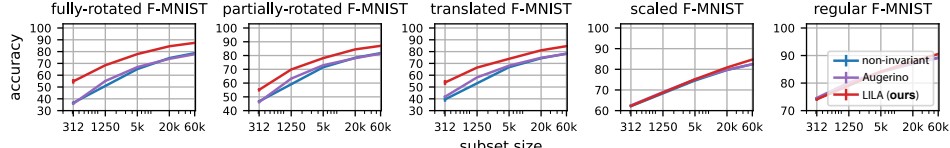

Figure 19: CNN on MNIST

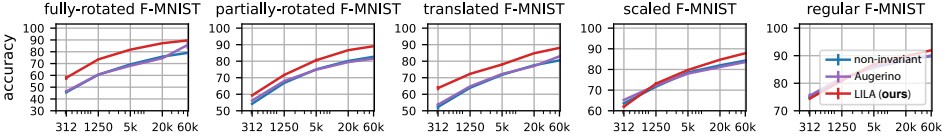

Figure 20: MLP on F-MNIST

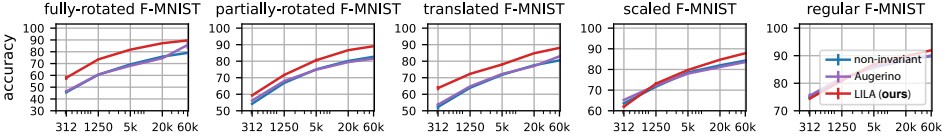

Figure 21: CNN on F-MNIST