# OpenReview forum: "Invariance Learning in Deep Neural Networks with Differentiable Laplace Approximations"
_NeurIPS.cc/2022/Conference — NeurIPS 2022 Accept_

### Official Review · Reviewer_LhTL · 2022-07-11

**Rating:** 7
**Confidence:** 2
**Soundness:** 4 excellent
**Presentation:** 3 good
**Contribution:** 3 good

**Summary:**

The authors extend previously proposed techniques for a Laplace approximation for Bayesian model selection to learn and incorporate invariances while training a deep network. To make this approximation computationally feasible, the authors propose techniques to apply a Kronecker-Factored approximation. The authors extensively evaluate their method on many datasets, with ablations and comparisons to another method that learns invariances while training.

**Questions:**

* Does the baseline get trained with any augmentations applied?
* Have the authors considered using a generative model in place of the six hand-picked invariances (e.g., a disentangled variational auto-encoder like the Beta-VAE from Higgins et al 2011)? This would take the work one step closer to true a priori invariance learning.

**Limitations:**

* From the experiments the authors have presented, it appears that their methods requires the set of candidate invariances to be specified beforehand. This is limited for datasets where the invariances are not known beforehand.
* The authors have not released code.

**Strengths And Weaknesses:**

Strengths
* The authors have very illustrative experiments (Figure 2 is very well-presented) that show that the models they train are learning the invariances present in the dataset. Training a network to capture the invariances present in a dataset is of significant interest for the machine learning community. Furthermore, they test their method on a wide variety of datasets/network architectures.
* The authors clearly describe the novelty of their approach in comparison to Immer et al 2021. They describe how the aforementioned work would be computationally infeasible for their proposed task.

Weaknesses
* One goal of this work is “obtain a model that is invariant to a set or distribution of transformations” (lines 16-17) as measured by test accuracy in Table 1. In the case where one wants s network with invariance to the natural variations in a dataset, the authors should justify their approach over simply training with augmented samples, as done in (RandAugment, Cubuk et al. 2019). For example, can the authors’ approach provide benefit over RandAugment on ImageNet using their six candidate augmentations? It should be stated that the authors’ proposed method has significance outside this scenario, since they also identify which invariances are present (as opposed to augmentation approaches). Nevertheless, such a comparison is important for the community to understand which method should be used for this given circumstance.
* Figure 4 of the appendix is very nice for measuring performance of different approximation methods. To strengthen their work, the authors should consider giving empirical runtime results as well so that practitioners can understand the empirical performance/run speed tradeoffs.

---

> ### Author Response · Authors · 2022-08-01
> **Author Response**
>
> Thank you for your feedback and help in improving the paper. We address your concerns below and are happy to answer follow-up questions or discuss further.
>
> > justification over RandAugment (Cubuk 2019)
>
> In RandAugment, the authors "postulate that a single global distortion M may suffice for parameterising all transformations" and tie various invariances together. The approach would therefore not work on data sets where only individual invariances are present, e.g., partially rotated data sets. As correctly pointed out by the reviewer, our method can further identify individual invariances, e.g. scale or rotation, independently of others.
>
> We also would like to emphasise that the proposed method to learn invariances using gradients is applicable when little or no validation data are available, e.g., in interactive learning scenarios. Unlike Augerino, our approach does not require additional hyperparameters.
>
> > the authors should consider giving empirical runtime results as well
>
> Thanks for the suggestion. We will include runtimes in the appendix.
>
> > does the baseline get trained with any augmentations applied?
>
> The baseline is trained without augmentations because we assume invariances are unknown a priori. Using validation data, we could consider choosing data augmentation with cross-validation as a baseline. However, this quickly becomes infeasible for multiple hyperparameters. In our case, to assess only three discrete settings for each $\eta_i$ this would already require training $3^6=729$ models from scratch, which is why we did not include cross-validation as a baseline.
>
> > set of candidate invariances to be specified beforehand.
>
> The approach is general in that it can work for any group or set of transformations, as long as it can be differentiably parameterised. Learning invariances from within all affine transformations is currently commonly used (van der Wilk 2018, Benton 2021). Parameterising distributions over other groups is an interesting avenue for future work, but orthogonal to enabling differentiably learning such parameterisations.
>
> > have the authors considered using a generative model for invariances?
>
> This is certainly an interesting idea. The focus and main contributions of the paper are to allow optimising parameterisable invariances in a way that scales to deep neural networks. Affine invariances, as commonly used in this context, allow us to validate the performance of our model by inspecting learned $\eta$ on transformed datasets. Our method can be applied to any generator, while Augerino does not work in this case as it relies on a heuristic regulariser. This would be an interesting application of our method, but orthogonal to enabling differentiably learning of such parameterisations.
>
> > the authors have not released code.
>
> We will publish code upon acceptance.

---

### Official Review · Reviewer_YcvR · 2022-07-11

**Rating:** 7
**Confidence:** 3
**Soundness:** 2 fair
**Presentation:** 3 good
**Contribution:** 2 fair

**Summary:**

The paper proposes a gradient-based method for automatically selecting the data augmentation parameters without the validation data. The trick is to parameterize the data augmentation scheme, approximate the marginal likelihood with respect to these data augmentation parameters and optimize them during training. The authors propose schemes such as KFAC approximation and explicit automatic differentiation to make the computation of differentiable marginal likelihood more tractable. Empirically, the proposed approach is capable of recovering invariances presented in the data on MNIST, FashionMNIST, and CIFAR10.

**Questions:**

Comments:
* In the introduction, the authors state that choosing the correct transformation is expensive and requires trial and error. However, the proposed approach also requires selecting the parameterization of $\eta$, which also requires some prior knowledge of the dataset and task. For example, in the case of image classification, affine invariance parameterization looks like a reasonable choice, but how should we select the parameterization for other tasks (e.g., language modelling). It would be helpful if the authors addressed these issues.
* In Table 2, is the model evaluated on the standard CIFAR-10 datasets, or is it a corrupted version (e.g., scaled and translated)? The test accuracy for the baseline method seems to be a little lower than I expected.
* As the paper introduces a lot of notations, it would be helpful to have a table of notations summarizing all notations used in the paper in the Appendix.
* As the authors mentioned in the checklist, releasing the code would be helpful.
* What additional hyperparameters does the proposed method require, and how sensitive are they? Given a new dataset, can the proposed algorithm work without putting a lot of effort into tuning these hyperparameters?

Minor Comments:
* Some colours in Figure 2 (e.g. vertical & horizontal scaling) are too similar, and it is difficult to distinguish them in the plot when printed.
* In line 301, there is a missing bracket after App. B.

**Limitations:**

The work does not have a potential negative societal impact.

**Strengths And Weaknesses:**

Strengths:
* The topic is interesting and relevant to the Neurips community.
* The paper is well-written, and the technical details look correct to me.
* While limited to small datasets and models, the experiments demonstrate that the proposed approach can successfully recover invariances presented in the model and can help improve generalization.

Weaknesses:
* There are several specific limitations of the proposed approach. To give some specific examples: (1) the proposed approach does not scale to large datasets and models, (2) the method requires task-specific data augmentation parameterization (e.g., affine transformation for image classification tasks), and it is not straightforward to apply the proposed method in other domains, and (3) the proposed algorithm is challenging to implement on standard deep learning frameworks and introduce additional computational overhead that scales quadratically with model parameters and linearly to the number of datasets.
* While I believe that the toy experiments presented in the paper demonstrate the proposed approach's effectiveness, the proposed method's scalability is still questionable.
* It would be helpful if the authors also compared their algorithms with other data augmentation adaptation methods that make use of validation datasets. To my understanding, these methods should be much more efficient than the proposed approach, although it requires a validation dataset. If we select 20% of training data as validation data, shouldn't the prior algorithms also be able to learn these invariances?


** Post-Rebuttal **

Thank you for the author's response. I acknowledge that I have read the rebuttal and other reviewers' comments. As the author addressed all my concerns in the paper, I increased my score.

---

> ### Author Response · Authors · 2022-08-01
> **Author Response**
>
> Thank you for your feedback and help in improving the paper. We address your concerns below and are happy to answer follow-up questions or discuss further.
>
> > (1) the proposed approach does not scale to large datasets and models
>
> Our proposed approach enables gradient-based invariance learning on deep ResNet architectures on large image classification datasets without validation data. This is a step change compared to prior work using Bayesian model selection that relies on Bayesian inference in single layers only (van der Wilk 2019; Schwöbel 2021; van der Ouderaa 2021). Our contributions enable scaling to even larger settings but would require additional engineering.
>
> > (2) the method requires task-specific data augmentation parameterisation and is not straightforward in other domains. [...] approach requires selecting the parameterisation
>
> The approach is general in that it can work for any group or set of transformations, as long as it can be differentiably parameterised, which is the case also in other domains. Learning invariances using training data only from within all affine transformations is still actively investigated (van der Wilk 2018, Benton 2021), and we provide concrete improvements. Parameterising distributions over other groups is an interesting avenue for future work, but orthogonal to enabling differentiably learning such parameterisations, which is what our paper focusses on.
>
> > (3) the algorithm is challenging to implement on standard deep learning frameworks and the computational overhead scales quadratically with model parameters
>
> To allow easy implementation, we phrase the gradient-estimator as a vector-Jacobian product, a standard tool of deep learning frameworks. While this is certainly more challenging to implement compared to a standard backward-pass, frameworks like `jax` and `functorch` gradually enable more elaborate methods. The computational overhead does not scale quadratically with model parameters but ranges from linearly ($\mathcal{O}(P)$) to sub-quadratically ($\mathcal{O}(P^{1.5})$) in parameters $P$ (App. C). In practice, the main computational overhead is, like for Augerino, linear in the number of augmentation samples. For example, future research on parameterising invariances without sampling could therefore make our method as fast as standard training.
>
> > comparison to algorithms that use validation data
>
> Indeed, validation data can also be used to learn invariances. However, this quickly becomes infeasible for multiple hyperparameters. In our case, to assess only three discrete settings for each $\eta_i$ would already require training $3^6=729$ models from scratch, which is why we did not include cross-validation as a baseline. The appeal of gradient-based joint optimisation, which we focus on, is that we can train invariance parameters in a *single*, albeit slightly more expensive, training run. We will make sure to discuss this in the paper.
>
> > what hyperparameters are required
>
> The main benefit and motivation of our method is that it does not require any additional hyperparameters, thereby reducing the numbers of hyperparameters that need to be chosen. Our method only requires setting a learning rate and standard settings with Adam work well. This is unlike Augerino, which requires designing and tuning a separate regulariser using a validation set.
>
> > is the model in Table 2 evaluated on the standard CIFAR-10 datasets?
>
> Yes, this is standard CIFAR-10. The results are on a rather small ResNet. We have since run the experiment with a WideResNet 16-4, a common model for CIFAR-10 that commonly achieves 85% accuracy without data augmentation, and show the results below.
> | baseline | augerino | ours  |
> |----------|----------|-------|
> | $85.17\pm0.39$ %   | $87.67\pm0.08$ % | $91.86\pm0.05$ % |
>
> > notation summary
>
> We think this is a great idea and will add this to the appendix.
>
> > code
>
> We will publish code upon acceptance.

---

### Official Review · Reviewer_Dhxp · 2022-07-11

**Rating:** 6
**Confidence:** 2
**Soundness:** 3 good
**Presentation:** 3 good
**Contribution:** 3 good

**Summary:**

The paper proposes to select parameters for augmentation through a training procedure (without validation data, like the prior work on Augerino). It does so by introducing a Bayesian model selection framework for selecting the ideal set of augmentations for a neural network. Effort in making this model tractable is notable and the evaluation shows some benefits over strong baselines.

**Questions:**

* What parts of the design of your model are entirely novel and what parts are adapted from prior work (e.g., how KFAC, Laplace-GGN, etc. fit in with prior work)?
* What are the computational concerns with this method? How does it compare with Augerino in terms of training and inference time?
* What is the significance of the baseline used in experiments?

**Strengths And Weaknesses:**

Strengths:
* The paper is clear and reads well, though the treatment of technical derivations is quite long.
* Model does sensible behavior (i.e., Figure 2).
* Model has improvements over Augerino (e.g., Table 1).
* Diversity of datasets, models, and augmentations seems acceptable, but can be still strengthened (e.g., Table 1).

Weaknesses:
* One baseline is non-invariant, which seems like a weak baseline, since there is often a significant performance boost by just using augmentations in a simple way. Augerino is a stronger baseline, but the results are much less noticable against that baseline (e.g., Figure 3).
* It's not clear nor intuitive why this method works. Some examples of failure modes of Augerino that were addressed would be enlightening.

Post Rebuttal:
Many of the paper's strengths are due to being a Bayesian treatment of the problem of setting augmentation parameters, which avoids hyperparameter tuning. There is a convincing case that it would be used in the future as a building block as this approach works on a standard set of benchmarks and is otherwise sensible. The evaluation can be strengthened (as always, e.g., scaling the approach on more difficult datasets), but I think the paper has sufficient contribution as is, and so I am increasing my score.

---

> ### Author Response · Authors · 2022-08-01
> **Author Response**
>
> Thank you for your feedback and help in improving the paper. We are happy to answer follow-up questions.
>
> > no baseline with augmentation in a simple way
>
> In our experiments, we assume that invariances are unknown a priori. Simple augmentation settings would therefore not work. We could consider choosing data augmentation with cross-validation as a baseline. However, besides requiring validation data, this quickly becomes infeasible for multiple hyperparameters. In our case, to assess only three discrete settings for each $\eta_i$ this would already require training $3^6=729$ models from scratch, which is why we did not include cross-validation as a baseline.
>
> > Some examples of failure modes of Augerino that were addressed
>
> There are two main failure modes that our method overcomes:
> 1. Augerino requires an additional hyperparameter that needs to be tuned. In our experiments, we find that this hyperparameter is non-trivial to tune and requires an additional validation set. Since our method follows from Bayesian model selection, there is no additional hyperparameter. Example: https://i.imgur.com/VcDSPHq.png
> 2. Augerino depends on the parameterisation of the invariance parameter $\eta$ as it uses the heuristic that increasing $\eta$ corresponds to increasing invariance. This makes it fail, for example, if the parameterisation of invariances changes to $1/\eta$. Our method can be applied without requiring knowledge about the particular parameterisation. Example: https://i.imgur.com/0ZsBFnn.png
>
> We will add these examples and other failure cases that we identified to the appendix to further strengthen the advantages of our proposed approach over existing methods.
>
> > it's not clear nor intuitive why this method works.
>
> Our method relies on the principle of Bayesian model selection that trades off performance and model complexity and is well-studied in statistics. We discuss this briefly in Sec. 3.2, but recommend the references therein. Bayesian model selection is also introduced in the common ML textbooks: Sec. 28 in MacKay’s Information Theory, Inference, and Learning Algorithms, Sec. 5.6 in Murphy’s ML book, and Secs. 3.4 and 4.4 in Bishop’s PRML book. Van der Wilk et al. (2018), referenced in the paper, discuss the underlying mechanism applied to invariance learning in the context of Gaussian processes, where Bayesian model selection is the standard.
>
> **Intuitively**, a model is *simpler* if it can explain two data points $x, x'$, where one is a transformed version of the other. A non-invariant model needs to fit both data points individually and is more complex. Our objective prefers the simpler model in this case and successfully identifies the invariance.
>
> **Mathematically**, the log-determinant part of the marginal likelihood approximation favours models that uncover invariances. Consider data points $x$ and transformed $x'$ to which we want to be invariant. Then, an invariant model would have identical Jacobians $J(x)$ and $J(x')$. This leads to a minimal log-determinant because the angle between the Jacobians is trivially zero.
>
> An **alternative explanation** is that our method adjusts the invariance to obtain a model with a *flatter minimum*. In the deep learning community, it is widely hypothesised that flatter minima generalise better, as has been discussed by Hochreiter and Schmidhuber.
>
> We will elaborate on these aspects to make the mechanisms of the method clearer.
>
> > novelty
>
> Previous work on Bayesian model selection (e.g., Immer et al. 2021) optimises only simple hyperparameters that do not fundamentally change the inductive bias of the network, e.g., weight decay parameters. Our contributions (Sec. 4) enable to learn more complex inductive biases that act on the neural network directly, like invariances. To achieve this, we extend the Laplace-GGN and KFAC approximations to invariant neural networks (Secs. 4.1 and 4.2) and propose a method that enables gradient-based optimisation of complex hyperparameters without memory overhead (Sec. 4.3). Our innovations enable Bayesian model selection in neural networks for complex hyperparameters for the first time, which we demonstrate for learning invariances.
>
> > computational concerns
>
> We discuss computational complexity in detail in Appendix C and will add runtime benchmarks. In practice, the main increase in computational cost is by the same constant factor as existing methods (Augerino) for both training and inference. This is the number of augmentation samples. Our method provides clear benefits over Augerino, i.e., better performance, no additional hyperparameter, and overcoming the failure modes discussed earlier.
>
> > what is the significance of the baseline used in experiments?
>
> We include the non-invariant model to emphasise the gain of performance and data-efficiency when learning invariances.

---

> > ### Author Response · Authors · 2022-08-08
> > **Discussion Reminder**
> >
> > Dear reviewer Dhxp,
> >
> > We would appreciate a timely response to our rebuttal so that we can still address follow-up questions or concerns you might have.
> >
> > Sincerely,
> >
> > Authors

---

> > > ### Comment · Reviewer_Dhxp · 2022-08-08
> > > **Response**
> > >
> > > Thank you for the thoughtful reply. I have updated my review.

---

### Author Response · Authors · 2022-08-01
**General Author Response**

We would like to thank all reviewers for their time and effort in helping us improve the paper.

We are happy that the overall sentiment is positive and the reviewers think our work is of interest to the NeurIPS and ML community (YcvR, LhTL), that the method provides clear benefit over Augerino (Dhxp, YcvR), and that the paper is well-written and illustrated (all reviewers). No reviewer found any factual mistakes in our paper.

The main concern brought up by all reviewers is scalability, which is a valid concern that affects all prior work in similar ways, such as Augerino and methods for Bayesian model selection of invariance. However, we would like to emphasise that our method is the **first to enable scaling Bayesian model selection to learn complex hyperparameters, like invariances, in deep neural networks**, which has so far only been successful for single-layer models or Gaussian processes. Our work thus affirms that Bayesian model selection can be useful for selecting complex inductive biases. There are at least two reasons we believe such research is important:
1. Invariances and complex inductive biases have parameterisations with a dimensionality that makes grid search cumbersome. Gradient-based optimisation helps to avoid suffering from the curse of dimensionality. Our work is a step towards this by enabling differentiation of the marginal likelihood w.r.t. complex hyperparameters and learning them together with the weights of a neural network.
2. When data are scarce, for example in interactive scenarios like active learning, validation-based approaches do not work or have high variance. Bayesian model selection allows to train hyperparameters jointly on the training data and therefore has the potential to be useful in such scenarios.

We believe that pushing in this direction is of strong relevance to the community (as also recognised by reviewers YcvR and LhTL).

We further address the remaining concerns and questions of all reviewers in individual responses.

---

### Meta-Review · Area_Chair_M5nK · 2022-08-26

**Recommendation:** Accept
**Confidence:** Certain

**Metareview:**

After reading the submission and its reviews, my understanding is that the submission proposes to use a tractable Laplace approximations (General Gauss-Newton and K-FAC) to learn suitable invariances/augmentation for the considered neural network. The derivations of this paper and the description of the method are clear and well motivated. While the experiments illustrate the concept and the soundness of the method, they remains small scale and are limited to sets of parameterized augmentations. Nonetheless, I recommend this submission for acceptance.

**Award:**

No

---

### Decision · Program_Chairs · 2022-09-14

Accept